# The Ca$^{2+}$-activated chloride channel anoctamin-2 mediates spike-frequency adaptation and regulates sensory transmission in thalamocortical neurons

Go Eun Ha[1,*], Jaekwang Lee[2,*], Hankyul Kwak[1], Kiyeong Song[1], Jea Kwon[2], Soon-Young Jung[2], Joohyeon Hong[1], Gyeong-Eon Chang[1], Eun Mi Hwang[3], Hee-Sup Shin[4], C. Justin Lee[2] & Eunji Cheong[1]

Neuronal firing patterns, which are crucial for determining the nature of encoded information, have been widely studied; however, the molecular identity and cellular mechanisms of spike-frequency adaptation are still not fully understood. Here we show that spike-frequency adaptation in thalamocortical (TC) neurons is mediated by the Ca$^{2+}$-activated Cl$^-$ channel (CACC) anoctamin-2 (ANO2). Knockdown of ANO2 in TC neurons results in significantly reduced spike-frequency adaptation along with increased tonic spiking. Moreover, thalamus-specific knockdown of ANO2 increases visceral pain responses. These results indicate that ANO2 contributes to reductions in spike generation in highly activated TC neurons and thereby restricts persistent information transmission.

[1] Department of Biotechnology, College of Life Science and Biotechnology, Yonsei University, Seoul 03722, Republic of Korea. [2] Center for Neural Science, Korea Institute of Science and Technology, Seoul 02792, Republic of Korea. [3] Center for Functional Connectomics, Korea Institute of Science and Technology, Seoul 02792, Republic of Korea. [4] Center for Cognition and Sociality, Institute for Basic Science, Daejeon 34141, Republic of Korea. * These authors contributed equally to this work. Correspondence and requests for materials should be addressed to E.C. (email: eunjicheong@yonsei.ac.kr).

The firing patterns of neurons dynamically encode corresponding information in brain circuits[1] and are determined by the repertoire of ion channels expressed in each neuron. A reduction in the firing frequency of the spike response, known as spike-frequency adaptation, has been observed in various types of neurons in the thalamocortical (TC) neurons in the ventrobasal (VB) nuclei[2,3], hippocampal pyramidal neurons[4,5], the amygdala[6] and the cortex[7]. However, the specific ion channels and the molecular mechanisms of spike-frequency adaptation have not been elucidated.

TC neurons display firing patterns that reflect sensory information transmission from the thalamus to the cortex[8]. The relay of sensory information from the periphery to the cortex is a dynamic process involving modulation of information that is dependent on both the state of the thalamus and inputs from other brain regions[8,9]. The intrathalamic network[10,11] is composed of glutamatergic TC neurons and GABAergic thalamic reticular nucleus (TRN) neurons, as well as projections from other brain regions[12,13]. TC neurons integrate information from ascending sensory inputs, as well as projections from the TRN and other brain regions and transmit information to the cortex; this process is well established as one of the most efficient and reliable brain projection systems for driving cortical neurons[14,15]. Therefore, the firing rates and patterns of TC neurons determine the nature of information processed in the TC circuits.

Signal transmission from the thalamus to the cortex is reflected in two distinct TC neuron-firing patterns: tonic and low-threshold burst firing[13]. Tonic firing is generally accepted as a relay mode for sending afferent sensory signals to the cortex[13], while the exact role of burst firing with respect to sensory gating is still debated[16–19]. Numerous studies have demonstrated increases in cortical responses proportional to increases in TC tonic spikes[12,13,20], supporting the notion that tonic spikes are a strong indicator of the amount of sensory information relayed. TC neurons generate tonic spikes at comparatively regular intervals at low frequency; however, they display patterns with gradual increases in interspike intervals (ISIs) when hyper-activated by depolarization[3]. This form of activity-dependent spike-frequency adaptation is hypothesized as a mechanism for neuronal self-inhibition.

Spike-frequency adaptation in neurons is associated with slow-type afterhyperpolarization (AHP) currents, which can be further categorized into medium AHP (mAHP) and very slow AHP currents ($mI_{AHP}$ and $sI_{AHP}$), the decay kinetics of which are approximately hundreds of milliseconds and over seconds, respectively[21]. Of these two types of currents, mAHP is known to be mediated by small conductance (SK) or large conductance (BK) $Ca^{2+}$-activated $K^+$ channels in many types of neurons, including hippocampal and cerebellar neurons[5,22,23]. In particular, SK channels have been reported to mediate $Ca^{2+}$-activated AHP currents activated by T-type $Ca^{2+}$ channels in inhibitory TRN neurons in the thalamus[24]. Although the source of $Ca^{2+}$ that activates mAHP currents in TC neurons has been previously reported to be either $Ca^{2+}$ influx via voltage-gated $Ca^{2+}$ channels or intracellular $Ca^{2+}$ release[2,25], the channels involved in $Ca^{2+}$-activated mAHP currents and their role in TC neurons are not well characterized. Moreover, the functionality of these channels in thalamic sensory information processing has not been studied.

We demonstrate that the $Ca^{2+}$-activated $Cl^-$ channel (CACC), anoctamin-2 (ANO2), mediates spike-frequency adaptation in TC neurons. Knockdown of ANO2 markedly diminished the prolongation of ISIs and significantly decreased mAHP currents in TC neurons. Furthermore, thalamus-specific ANO2 knockdown significantly increased visceral pain responses. These results emphasize a critical role for ANO2 in preventing excessive spike generation in TC neurons and, therefore, in regulating sensory information relayed from the thalamus to the cortex.

## Results

**Spike-frequency adaptation in TC neurons is $Ca^{2+}$-dependent.** To characterize the firing pattern of TC neurons, we performed whole-cell patch clamping of individual TC neurons from VB nuclei, administering multiple depolarizing steps. Injection of a depolarizing 200 pA current induced tonic firing with gradually increasing ISIs in artificial cerebrospinal fluid (aCSF) containing 2.4 mM $[Ca^{2+}]_{ex}$ (Fig. 1a) as well as in that with physiological 1.8 mM $[Ca^{2+}]_{ex}$ (Supplementary Fig. 1a,b and Supplementary Note 1). The membrane potential was hyperpolarized compared with the resting potential (dotted line) after the depolarizing step was terminated (Fig. 1a). The subsequent replacement of extracellular buffer with $Ca^{2+}$-free buffer increased tonic-firing rates and simultaneously abolished spike-frequency adaptation both at 25 and 32 °C (Fig. 1a, Supplementary Fig. 1c and Supplementary Note 1). This was accompanied by a reduction in mAHP amplitude at the end of stimulation. In the absence of extracellular $Ca^{2+}$, the magnitude of AHP decreased by ~55% ($-4.72 \pm 0.44$ to $-2.10 \pm 0.37$ mV, $P < 0.001$ by paired $t$-test, $n = 10$ from five mice; Fig. 1b). Concomitantly, the number of tonic spikes in TC neurons increased by ~32% ($20.2 \pm 2.1$ to $26.6 \pm 1.9$, $P < 0.001$ by paired $t$-test, $n = 10$ from five mice; in response to 200 pA depolarizing currents, Fig. 1c). The adaptation index (AI) was obtained by dividing the average of the first two ISIs by that of the last two ISIs. The difference between AI values of TC neurons in $Ca^{2+}$-containing buffer ($0.59 \pm 0.04$) and $Ca^{2+}$-free buffer ($0.85 \pm 0.03$) suggested that $Ca^{2+}$ is essential for spike-frequency adaptation ($P < 0.001$ by paired $t$-test, $n = 10$ from five mice; Fig. 1d and Supplementary Fig. 1c,d). The input–output curve showed a voltage-dependent increase in the firing rate ($n = 10$ from five mice; Fig. 1e) similar to that of older mice[26] (Supplementary Fig. 2 and Supplementary Note 2). Together, these results suggest that both spike adaptation and mAHP in TC neurons are regulated by a $Ca^{2+}$-dependent mechanism.

In order to characterize the $Ca^{2+}$-activated mAHP current, the tail current following voltage steps to induce $Ca^{2+}$ influx was measured in the presence of extracellular tetrodotoxin (TTX). Strikingly, the observed tail current was a net inward current at $V_m$ of $-60$ mV, and its reversal potential was $\sim -55$ mV (Fig. 1f,g). This tail current was completely abolished when extracellular $Ca^{2+}$ was removed (Fig. 1h), which confirms $Ca^{2+}$ dependency. However, the observed net inward current is contrary to the expectation that $Ca^{2+}$-activated AHP currents are mediated solely by $K^+$ channels, which should generate an outward current at $V_m$ of $-60$ mV with $E_K$ near $-97.3$ mV in the intrapipette recording solution. To further investigate the possibility of components other than $K^+$ channels contributing to the tail current, we subsequently used a high $Cl^-$ (60 mM) intrapipette solution; this easily distinguished $K^+$ from anion components at $-60$ mV holding potential by maintaining the $E_{Cl}$ at approximately $-20$ mV and the $E_K$ at approximately $-97.3$ mV (Fig. 1i). Prepulse steps from $-60$ to $0$ mV could activate voltage-dependent $Ca^{2+}$ channels to induce the $Ca^{2+}$ influx into the cell. We tested prepulse steps to $0$ mV at various durations ranging from 50 ms to 1 s (Fig. 1i) and normalized the current decay time constant ($\tau$) of the slow component of each tail current to the mean $\tau$ ($32.1 \pm 7.1$ ms) induced by 50 ms prepulses. Longer prepulses, which are known to elicit larger $Ca^{2+}$ influxes, generated longer-lasting tail currents (Fig. 1j, $n = 9$ from four mice).

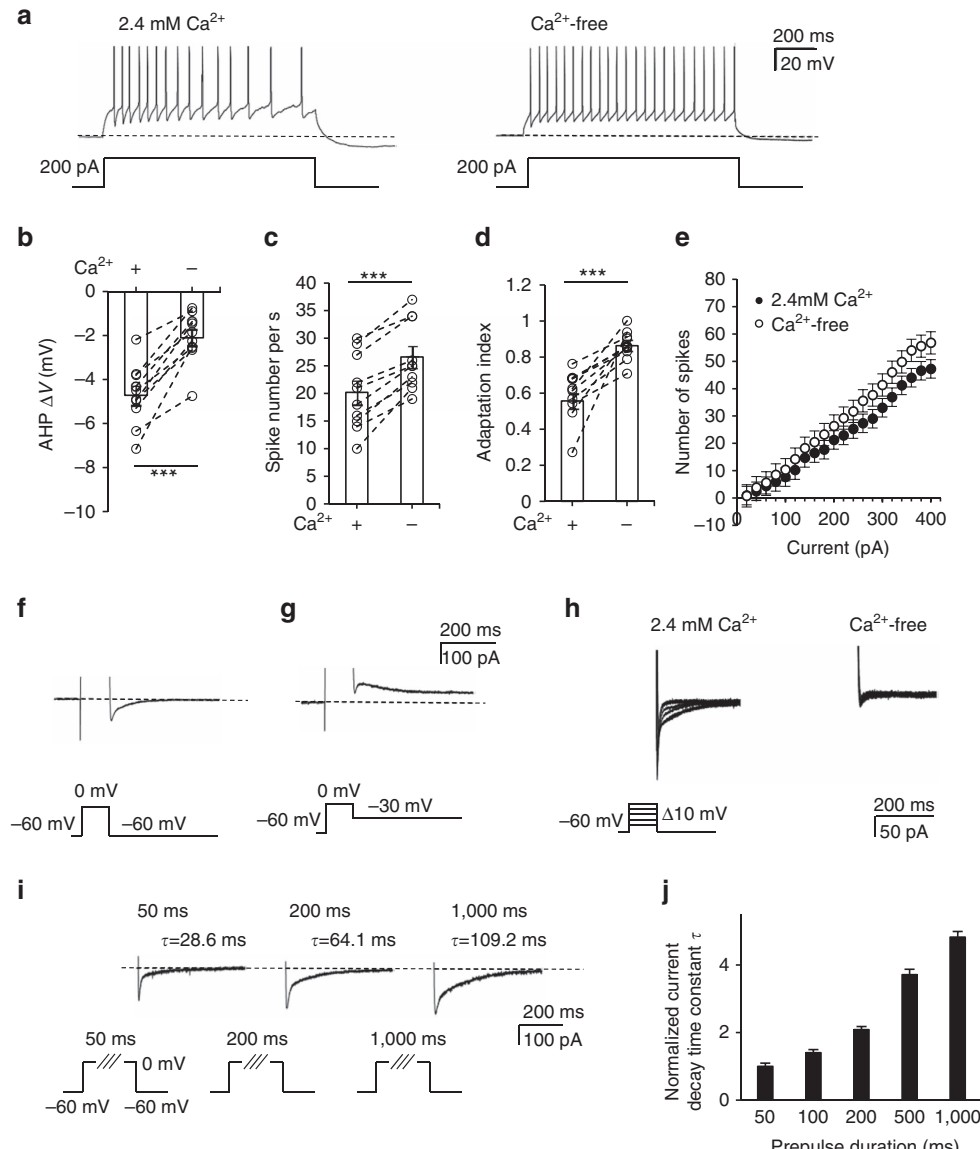

**Figure 1 | Extracellular $Ca^{2+}$ induces AHP and spike adaptation in TC neurons.** (**a**) A depolarizing current of 200 pA was injected for 1 s to induce tonic firing in TC neurons in 2.4 mM $Ca^{2+}$ buffer followed by replacement with $Ca^{2+}$-free buffer. (**b**) AHP potential after current injection averaged in both 2.4 mM $Ca^{2+}$ buffer and $Ca^{2+}$-free buffer ($n=10$ from five mice). A significant decrease in post-current injection AHP was observed in $Ca^{2+}$-free buffer compared with 2.4 mM $Ca^{2+}$ buffer (\*\*\*$P=0.0002$, paired $t$-test). (**c**) The number of spikes during current injection increased with a change from 2.4 mM $Ca^{2+}$ buffer to $Ca^{2+}$-free buffer (\*\*\*$P=0.0001$, paired $t$-test), and (**d**) was accompanied by an increase in the adaptation index ($n=10$ from five mice; \*\*\*$P=0.0005$, paired $t$-test). (**e**) The input–output (I–O) curve showed the number of spikes as measured under various current injections. Firing rates were increased in $Ca^{2+}$-free buffer compared with 2.4 mM $Ca^{2+}$ buffer ($n=10$ from five mice). (**f,g**) Representative traces of tail currents after 100 ms of a 0 mV depolarizing step protocol demonstrated a net inward current (**f**) at a $-60$ mV holding potential, which became an outward current at a $-30$ mV holding potential (**g**). (**h**) $Ca^{2+}$-activated tail currents were recorded in TC neurons in 2.4 mM $Ca^{2+}$ buffer followed by replacement with $Ca^{2+}$-free buffer in the presence of extracellular TTX. Under voltage clamp, the holding potential was $-60$ mV, and 100 ms voltage steps ranging from $-50$ to 0 mV ($+10$ mV per step) were administered. Tail currents were almost completely abolished in $Ca^{2+}$-free buffer compared with 2.4 mM $Ca^{2+}$ buffer. (**i,j**) The tail currents following voltage steps to 0 mV for durations of 50, 100, 200, 500 and 1,000 ms were measured in the presence of extracellular TTX. (**i**) The tail currents obtained from 50, 200 and 1,000 ms are shown. (**j**) The current decay constant obtained from each step was normalized to the current decay constant recorded over a 50 ms prepulse ($n=9$ from four mice). Longer prepulses produced longer current decay times. Error bars represent mean ± s.e.

These results suggest that spike-frequency adaptation patterns and mAHP currents in TC neurons are activated by the $Ca^{2+}$ influx from the extracellular space. The net inward tail currents at a $V_m$ of $-60$ mV indicated that $K^+$ is not a major contributor to the tail currents. Therefore, we further characterized AHP currents.

**$Ca^{2+}$-activated tail currents in TC neurons are primarily $Cl^-$.** Next, the components of the tail current were pharmacologically dissected (Fig. 2a,b). We first determined the contribution to the tail current by SK channels, which are a major component of AHPs in TRN neurons[24], or BK channels[22]. The amplitude of the inward tail current was increased by apamin, a selective SK channel blocker, indicating that SK current was masked by a large

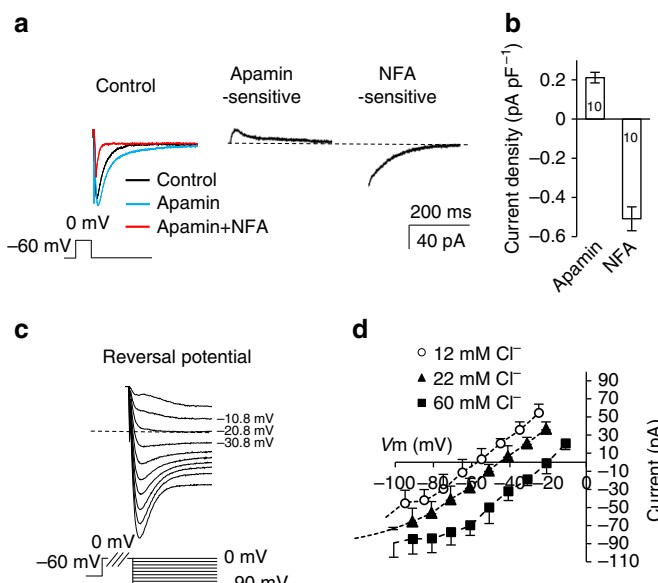

**Figure 2 | The components of Ca²⁺-activated AHP currents.**
(**a**) Representative traces of AHP currents after 50 ms of 0 mV depolarizing steps. To distinguish K⁺ and anion components, a 133 mM [K⁺]_in and 60 mM [Cl⁻]_in intrapipette solution was used. Ca²⁺-activated currents were recorded after application of TTX (black), TTX + apamin (blue) and TTX + apamin + NFA (red). Note that the apamin-sensitive K⁺ current is an outward current, and the NFA-sensitive current is inwards. (**b**) Current densities for each dissected current are shown ($n = 10$ from seven mice). (**c,d**) Reversal potentials of anion currents at various [Cl⁻]_in concentrations were measured with apamin-containing extracellular buffer. Reversal potentials changed along with [Cl⁻]_in content, indicating that Ca²⁺-activated anion currents are primarily composed of Cl⁻. Error bars represent mean ± s.e.

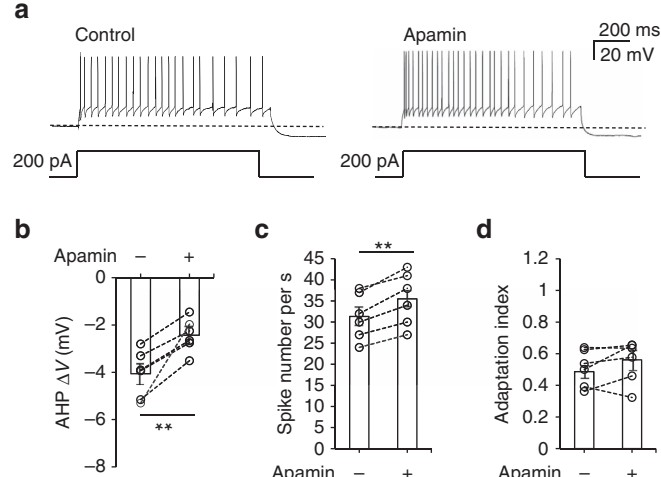

**Figure 3 | SK channels conduct mAHP currents but do not affect spike-frequency adaptation in TC neurons.** (**a**) A 200 pA depolarizing current was injected for 1 s to induce tonic firing in a TC neuron in apamin-free buffer followed by apamin-containing buffer. (**b**) AHP potentials after current injection were averaged in both apamin-free and apamin-containing buffer ($n = 6$ from three mice; $^{**}P = 0.0054$, paired t-test). A substantial decrease in post-injection AHP was observed when apamin was applied. (**c**) The number of spikes during current injection was increased by apamin treatment ($n = 6$ from three mice; $^{**}P = 0.0096$, paired t-test). (**d**) No change in the adaptation index was observed ($n = 6$ from three mice), indicating that SK channels do not significantly contribute to spike adaptation in TC neurons ($P = 0.25$, paired t-test). Error bars represent mean ± s.e.

inward tail current. Subsequently, the remaining current was almost completely blocked by the non-selective anion channel blocker niflumic acid (NFA) at a concentration of 50 μM (Fig. 2a). The apamin-sensitive SK current and NFA-sensitive current were obtained by trace subtraction (Fig. 2a). A selective blocker for BK channels, iberiotoxin, did not affect the Ca²⁺-activated tail current or tonic-firing frequency adaptation. The current densities of apamin-sensitive SK currents were less than a half of those of NFA-sensitive currents (apamin: $0.21 \pm 0.03$ pA pF⁻¹, NFA: $0.51 \pm 0.06$ pA pF⁻¹, $n = 10$ from seven mice; Fig. 2b). These two components contributed to the majority of the total tail current (Fig. 2a).

In order to verify whether NFA-sensitive currents were mediated by Cl⁻ channels, we measured the reversal potential of inward tail currents using intrapipette solutions containing various Cl⁻ concentrations in the presence of extracellular TTX and apamin (Fig. 2c). Junction potential correction was applied to curves I–V for NFA-sensitive inward currents obtained with intrapipette solutions containing 12, 22 and 60 mM Cl⁻ (Fig. 2d). The reversal potentials identified by curves I–V were $-55.8$, $-42.5$ and $-18.3$ mV, which were similar to the calculated values of $-61.5$, $-45.93$ and $-20.2$ mV, respectively. These results indicate that the inward tail current was conducted via Cl⁻ channels.

Following the finding that Ca²⁺-activated tail currents were mediated by the SK channel and an unidentified CACC, we examined the contribution of each of these channels to spike-frequency adaptation and the Ca²⁺-activated AHP currents described in Fig. 1. We were able to explore the role of the SK channel in TC neurons, since a selective SK blocker was available.

**SK channels do not affect spike-frequency adaptation**. We tested whether SK channels conduct AHP currents and develop spike adaptation in TC neurons. A depolarizing current (200 pA) applied to TC neurons induced tonic firing, followed by a long-lasting hyperpolarization of the membrane potential (Fig. 3a). Subsequent bath application of apamin substantially reduced AHP amplitudes (Fig. 3a) by ∼39% ($-4.1 \pm 0.41$ versus $-2.5 \pm 0.29$ mV, $P < 0.01$ by paired t-test, $n = 6$ from three mice; Fig. 3b) and increased the tonic-firing rate by ∼13% ($31.3 \pm 2.25$ versus $35.5 \pm 2.57$, $P < 0.01$ by paired t-test, $n = 6$ from three mice; Fig. 3c). However, the spike-frequency adaptation pattern, characterized by gradual prolongation of ISIs was not altered (Fig. 3a). This finding was confirmed by the absence of a significant change in the AI ($0.46 \pm 0.04$ versus $0.52 \pm 0.06$, $P = 0.22$ by paired t-test, $n = 6$ from three mice; Fig. 3d).

These results indicate that SK channels contribute substantially to the generation of Ca²⁺-activated mAHPs and tonic spike firing rate but are not critical to spike-frequency adaptation. Therefore, we predicted that spike-frequency adaptation and the remaining mAHP currents in TC neurons are mediated by CACCs.

**Cl⁻ currents with endogenous Cl⁻ confirm an inhibitory role**. We next sought to identity the CACC responsible for the observed effects and characterize the associated Cl⁻ currents in TC neurons. However, no previous study has described a CACC that is involved in mAHP currents and neuronal spike patterns in the central nervous system. Moreover, a limitation of the data described above was that all electrophysiological recordings were performed in a whole-cell configuration with an artificial source of intracellular Cl⁻. Therefore, we determined whether Cl⁻ currents generated by endogenous TC neuron ionic content would elicit hyperpolarization.

In order to avoid disruption of intracellular $Cl^-$, gramicidin-perforated patch recordings were performed. The equilibrium potential of GABA-induced, $GABA_A$-mediated current ($E_{GABA}$) in TC neurons was obtained at different holding potentials while puffing GABA at a distance of 50–100 μm from the soma of TC neurons, in the presence of 10 μM 3-aminopropyl (diethoxymethyl)phosphinic acid (CGP) 35348—a $GABA_BR$ antagonist. 6-cyano-7-nitroquinoxaline-2,3-dione (CNQX) and (2R)-amino-5-phosphonovaleric acid (APV) were also applied to block α-amino-3-hydroxy-5-methyl-4-isoxazole propionic acid and N-methyl-D-aspartate receptors, respectively (Fig. 4a, upper trace). The same experiment was repeated from the same cell after a whole-cell patch configuration was acquired (Fig. 4a, lower trace). $E_{GABA}$, shown as the intercept with the voltage axis on the I–V curve of the perforated-patch current (open circle), was $\sim -72$ mV ($n = 3$ from three mice) in TC neurons (Fig. 4b). Under symmetric $Cl^-$ concentrations between intracellular and extracellular buffer solutions, $E_{GABA}$, shown as the intercept of the I–V curve under the whole-cell configuration (solid circle), shifted to 0 mV ($n = 8$ from three mice), which strongly indicates that $E_{GABA}$ represents $E_{Cl}$ in TC neurons. The $[Cl^-]_{in}$ calculated from the measured $E_{GABA}$ was 8.4 mM. Since $GABA_A$ receptors, as well as most $CACCs^{27,28}$, are permeable to $HCO_3^-$ (ref. 29), this value was corrected with the relative permeability of 0.2 based on a 15 mM $[HCO_3^-]_{in}$ typical of central nervous system neurons[30]. The corrected endogenous $[Cl^-]_{in}$ was 5.4 mM. Thus, these results indicate that endogenous $Cl^-$ elicits inhibitory hyperpolarizing currents in TC neurons following action potentials.

**CACC expression in TC neurons.** We next investigated the molecular identity of the CACC-mediated spike-frequency adaptation in TC neurons. CACCs are expressed in various epithelia and astrocytes, as well as retinal, olfactory and sensory neurons[31–34]. On the basis of these previous studies, three CACCs were selected as feasible candidates for mediating $Ca^{2+}$-activated tail currents in TC neurons: bestrophin1 (BEST1), ANO1 and ANO2. The anoctamins, also known as transmembrane protein 16, are expressed in sensory and olfactory neurons[31,35], while the BEST1 channel is substantially expressed in the thalamus[34].

On the basis of this information, we first tested whether BEST1 was a feasible CACC candidate in TC neurons. Immunohistochemistry of thalamic sections using anti-BEST1 antibodies showed that the protein is highly expressed in VB nuclei in the TC region; however, it co-localized with the astrocytic marker glial fibrillary acidic protein (GFAP; Fig. 5a). Moreover, we found no difference in the NFA-sensitive current between $BEST1^{+/+}$ and $BEST1^{-/-}$ TC neurons ($0.51 \pm 0.06$ pA pF$^{-1}$, $n = 10$ from six mice and $0.42 \pm 0.10$ pA pF$^{-1}$, $n = 9$ from seven mice; $P = 0.40$ by t-test, Fig. 5b,c). These results indicate that the BEST1 channel is not the CACC mediating the observed tail current.

We next studied whether CACC currents were conducted by ANO1. Recently, we developed a potent ANO1 blocker, MONNA, which selectively blocks human ANO1 with an half-maximal inhibitory concentration ($IC_{50}$) of 1.27 μM. Furthermore, the compound does not block human BEST1, $Cl^-$ channel protein 2 or human cystic fibrosis transmembrane conductance regulator (hCFTR), even at concentrations of 10–30 μM (ref. 36). We examined whether MONNA could distinguish between ANO1 and ANO2 (Fig. 5d,g) via whole-cell current recordings from HEK293T cells expressing mouse ANO1 (mANO1) or mouse ANO2 (mANO2). The concentration-response curve demonstrating the blocking effect of MONNA on mANO1 at a $-100$ mV holding potential indicated an $IC_{50}$ of 1.19 μM, with nearly complete abolition of mANO1 currents at concentrations higher than 10 μM ($n = 6$; Fig. 5d,f). In contrast, mANO2 currents were blocked by only $\sim 20\%$ under identical conditions ($n = 8$; Fig. 5e,g). Therefore, we applied 30 μM MONNA to TC neurons at a $-100$ mV holding potential to determine whether it would block CACC currents in TC neurons (Fig. 5h). Application of MONNA blocked $\sim 20\%$ of CACC currents in TC neurons ($1.00 \pm 0.13$ versus $0.73 \pm 0.11$, $P < 0.05$ by paired t-test, $n = 5$ from three mice; Fig. 5i), which suggested two possibilities: ANO1 channels with low expression levels were blocked by MONNA or the $\sim 20\%$ blockade could be ascribed to MONNA blocking ANO2 channels. Regardless, these results indicate that ANO1 was not the primary mediator of CACC effects in TC neurons. Furthermore, data from western blot analysis and immunohistochemical staining confirmed that ANO1 is not expressed in the thalamus but is detected in kidney cells (Fig. 5j,k).

**The ANO2 channel is expressed in TC neurons.** To examine expression of ANO2 in TC neurons, we performed reverse transcription-PCR (RT–PCR) and western blot analysis. RT–PCR data indicated that ANO2 was expressed in the thalamic region, the olfactory epithelium[35] and mANO2-transfected NIH/3T3 cells, but not in control NIH/3T3 cells (Fig. 6a). The sequence of the RT–PCR product from thalamic tissue revealed a perfect match with that of the ANO2 sequence (NCBI Reference Sequence: NM_153589.2, Supplementary Fig. 3 and Supplementary Note 3). Western blot results demonstrated that ANO2 was expressed in the thalamic region and mANO2-overexpressing cells (Fig. 6b), consistent with the RT–PCR data. Immunohistochemical staining displayed ANO2 to be located on the membrane near the soma of TC neurons (Fig. 6c, Supplementary Fig. 4 and Supplementary Note 4).

An ANO2 blocker was not available; therefore, we developed an adeno-associated virus (AAV) vector (Supplementary Methods) expressing a small hairpin RNA (shRNA) to specifically

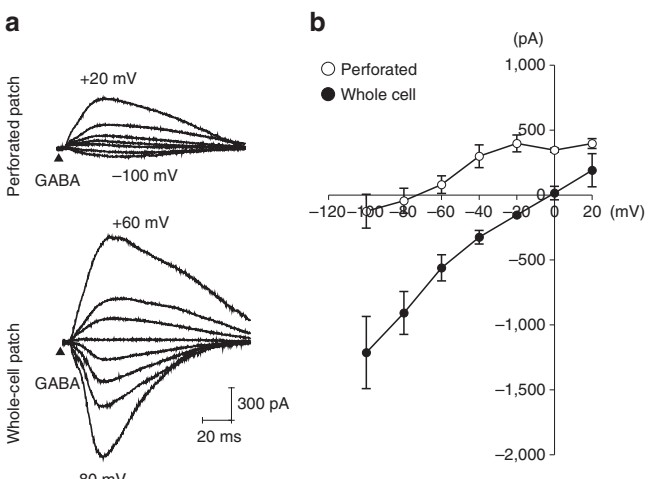

**Figure 4 | Determination of $E_{GABA}$ on TC neurons.** (**a**) Representative traces of a GABA response are shown at different holding potentials. In order to measure the equilibrium potential of the GABA ($E_{GABA}$) response, 1 mM GABA was administered with 20–30 p.s.i. pressure at a distance of 50–100 μm from the soma. To avoid disruption of $[Cl^-]_{in}$, a gramicidin-perforated patch recording was performed and compared with the whole-cell patch configuration (equimolar $[Cl^-]$ conditions both intracellularly and extracellularly). All recordings were performed in a bath solution containing 50 μM APV and 20 μM CNQX. (**b**) GABA responses at different potentials were plotted to show the I–V relationship. The intercept of the I–V curve of the perforated patch current (open circle, $n = 3$ from three mice) indicated that $E_{GABA}$ of TC neurons was $-72$ mV, whereas $E_{GABA}$ of the whole-cell current (solid circle, $n = 8$ from three mice) was 0 mV. Error bars represent mean ± s.e.

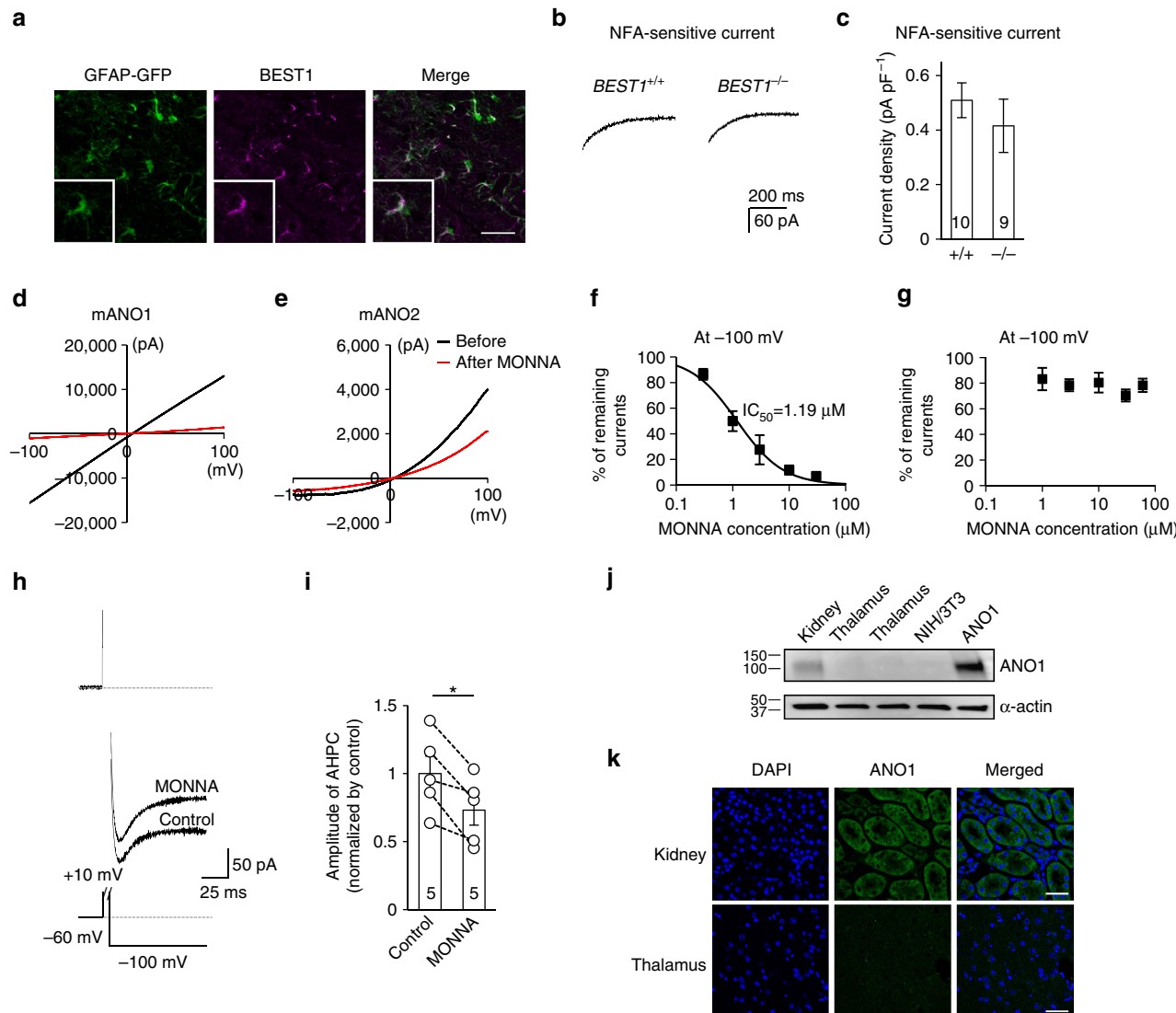

**Figure 5 | Ca$^{2+}$-activated Cl$^-$ channels in the thalamus.** (**a**) Immunostaining showed that the BEST1 channel (purple) was expressed in the thalamus and co-localized with the astrocytic marker GFAP (green; scale bar, 40 μm). (**b**) Representative trace of an NFA-sensitive current was obtained by subtracting the tail current in the presence of TTX and apamin from the tail current in the presence of TTX, apamin and NFA in wild-type (left trace) and *BEST1$^{-/-}$* (right trace) TC neurons. (**c**) Quantification of NFA-sensitive currents revealed that there were no differences between the NFA-sensitive currents of wild type ($n = 10$ from six mice) and *BEST$^{-/-}$* ($n = 9$ from seven mice) TC neurons ($P = 0.40$, *t*-test). (**d,e**) mANO1-IRES-EGFP or mANO2-EGFP-N1 was transfected into HEK293T cells. mANO1 and mANO2 currents were obtained by initially holding at 0 mV and then applying a 500 ms voltage ramp from −100 to +100 mV. Under whole-cell configuration, cells were treated with 30 μM MONNA and monitored until currents were blocked and stabilized. (**f**) Concentration-response curve at −100 mV indicated that the mANO1 current was almost completely abolished at high doses of MONNA ($n = 6$; 0.3, 1, 3, 10 and 30 μM shown); the IC$_{50}$ was 1.19 μM. (**g**) Concentration-response curve at −100 mV indicated that the mANO2 current remained largely intact even at high doses of MONNA ($n = 8$; 1, 3, 10, 30 and 60 μM shown). (**h**) A representative trace of AHP-induced current from a cell before and during 30-μM MONNA application for 20 min. The protocol to generate AHP-induced currents is described in the lower panel. The dotted line above indicates the baseline level. (**i**) Summary bar graph showing the effect of MONNA on the amplitude of AHP currents ($n = 5$ from three mice; *$P = 0.015$, paired *t*-test). (**j**) Western blotting data indicated that ANO1 channels were not expressed in the thalamus. (**k**) Immunostaining data showed that ANO1 (green) signal in the thalamus is very weak compared with that in kidney tissue, which was used as a positive control (scale bar, 20 μm). Error bars represent mean ± s.e.

knock down ANO2 and assess its function in TC neurons (Supplementary Fig. 5 and Supplementary Note 5). Mice injected with AAVs containing ANO2 shRNA2 (AAV-*shANO2*) or scrambled (AAV-*Scr*) vectors into the VB region displayed a red fluorescence signal within 10–11 days, confirming mCherry expression in TC neurons (Fig. 6d). Unlike AAV-*Scr*-infected neurons with intact ANO2 expression, AAV-*shANO2*-infected neurons had significantly reduced ANO2 expression ($72.68 \pm 4.65$, $n = 17$ versus $18.80 \pm 2.07$, $n = 16$, $P < 0.001$ by *t*-

test; Fig. 6d,e). Therefore, AAV-*shANO2* is suitable for studying functional changes in ANO2-deficient TC neurons.

**TC knockdown of ANO2 abolishes spike-frequency adaptation.** To test whether ANO2 conducts CACC currents in TC neurons, we performed patch clamp recordings of mCherry-positive neurons 10–14 days after injection of either AAV-*Scr* or AAV-*shANO2* into the VB region of 3-week-old

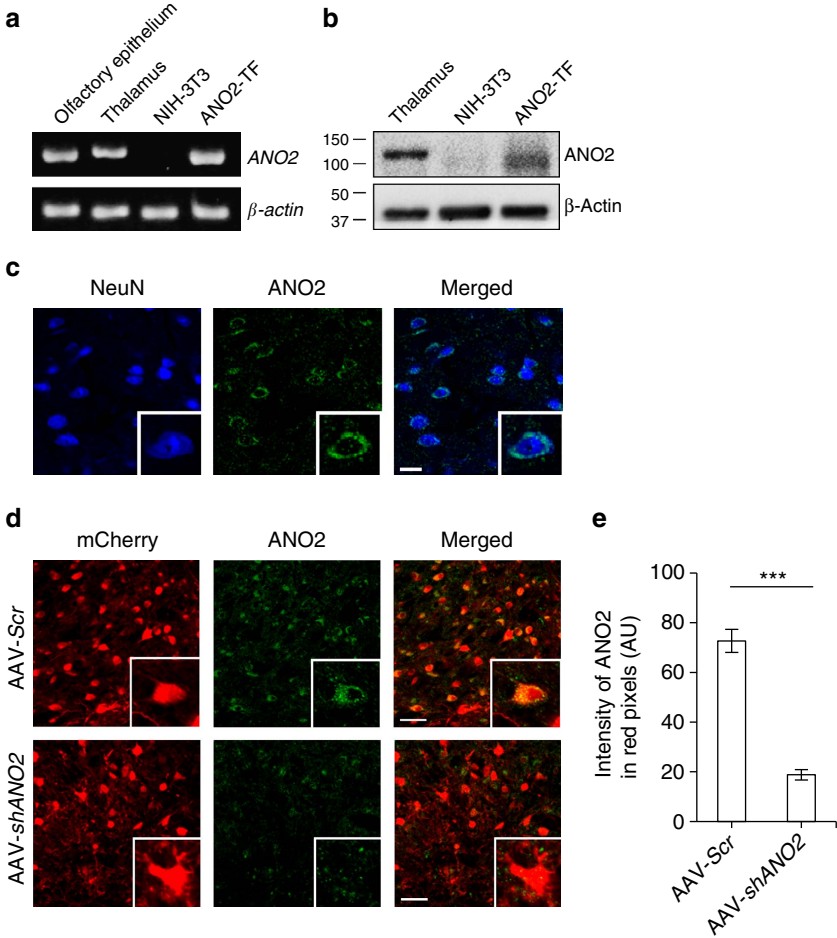

**Figure 6 | ANO2 channels expressed in the thalamus.** (**a**) RT–PCR data assessing *ANO2* expression demonstrated that *ANO2* was transcribed in the thalamus; *β-actin* was used as a reference transcript. (**b**) Western blotting data showed that ANO2 protein was expressed in the thalamus. ANO2-overexpressed cells displayed a broader ANO2 band related to various glycosylation levels, as described previously[35]. (**c**) Immunostaining data indicated that ANO2 (green) co-localized with the neuronal marker NeuN (blue), and that ANO2 was located in the soma of TC neurons (scale bar, 20 μm). (**d**) Immunostaining for ANO2 (green) showed that AAV-*Scr*-infected TC neurons (red, upper) clearly expressed ANO2 in the soma, but those infected with AAV-*shANO2* (red, lower) expressed ANO2 at a much lower level (scale bar, 50 μm). (**e**) Quantification of ANO2 expression in virus-infected TC neurons confirmed that gene silencing of ANO2 reduced the intensity of ANO2 expression in mCherry-positive neurons (AAV-*Scr* $n = 17$, AAV-*shANO2* $n = 16$, ***$P = 0.0000015$, *t*-test). Error bars represent mean ± s.e.

mice. A prominent reduction in the NFA-sensitive current was observed in TC neurons infected with AAV-*shANO2*, whereas those with AAV-*Scr* displayed intact NFA-sensitive currents ($0.15 \pm 0.02$ pA pF$^{-1}$, $n = 10$ from three mice versus $0.55 \pm 0.11$ pA pF$^{-1}$, $n = 6$ from six mice, $P < 0.05$ by *t*-test, Fig. 7a,b).

In conjunction with the reduction in NFA-sensitive currents, we observed a reduction in AHP following a depolarizing step in AAV-*shANO2*-infected TC neurons (Fig. 7c,d). Moreover, TC neurons transfected with AAV-*shANO2* showed higher tonic-firing rates and less spike accommodation than those transfected with AAV-*Scr* with depolarizing current injections (Fig. 7e,f) and EPSC-like stimulations (Supplementary Fig. 6, Supplementary Note 6 and Supplementary Methods). Knockdown of ANO2 reduced the magnitude of AHP elicited by current injection by $\sim 53\%$ ($-4.9 \pm 0.77$ mV, $n = 6$ from five mice versus $-2.3 \pm 0.28$ mV, $n = 7$ from five mice, $P < 0.01$ by *t*-test; Fig. 7d). This effect was accompanied by an $\sim 37\%$ increase in the tonic-firing rate ($25.2 \pm 2.02$, $n = 6$ from five mice versus $34.6 \pm 2.10$, $n = 7$ from five mice, $P < 0.01$ by *t*-test; Fig. 7e). Furthermore, spike-frequency adaptation was almost

completely eliminated, with an AI that increased by $\sim 55\%$ ($0.50 \pm 0.05$, $n = 6$ from five mice versus $0.77 \pm 0.03$, $n = 7$ from five mice, $P < 0.001$ by *t*-test; Fig. 7f). These results indicate that spike-frequency adaptation is primarily mediated by ANO2. To examine the contribution of ANO2 on single action potentials, we compared the second spike with the last spike in the train of tonic spikes, which reflects a time point when CACCs are expected to be active. There was no difference in either amplitude or rising kinetics of these action potentials (Fig. 7g). In addition, there was no difference in the half width of the action potential ($1.93 \pm 0.07$ ms, $n = 6$ from five mice versus $1.90 \pm 0.09$ ms, $n = 7$ from five mice, $P = 0.46$ by *t*-test; Fig. 7h), confirming that Ca$^{2+}$-activated AHP currents mediated by ANO2 channels do not significantly alter single action potentials.

In contrast, the pattern of low-threshold burst firing induced by the membrane potential rebound following the hyperpolarizing protocol was similar in AAV-*shANO2*-infected TC neurons and those infected with AAV-*Scr* (Fig. 7i). There was no difference in the number of intraburst spikes between AAV-*Scr*-infected and AAV-*shANO2*-infected TC neurons ($4.7 \pm 0.49$,

$n = 6$ from five mice versus $4.3 \pm 0.42$, $n = 7$ from five mice, $P = 57$ by $t$-test; Fig. 7j).

To further investigate the alteration of spike-frequency adaptation by ANO2 with intact $[Cl^-]_{in}$ concentration, we measured the spike pattern of TC neurons in the perforated patch configuration. AAV-shANO2-infected TC neurons showed less spike accommodation than AAV-Scr-infected TC neurons (Fig. 7k,l) similar to whole-cell patch clamp results. The spike-frequency AI was significantly increased by $\sim 71\%$ ($0.48 \pm 0.05$, $n = 6$ from four mice versus $0.28 \pm 0.06$, $n = 6$ from

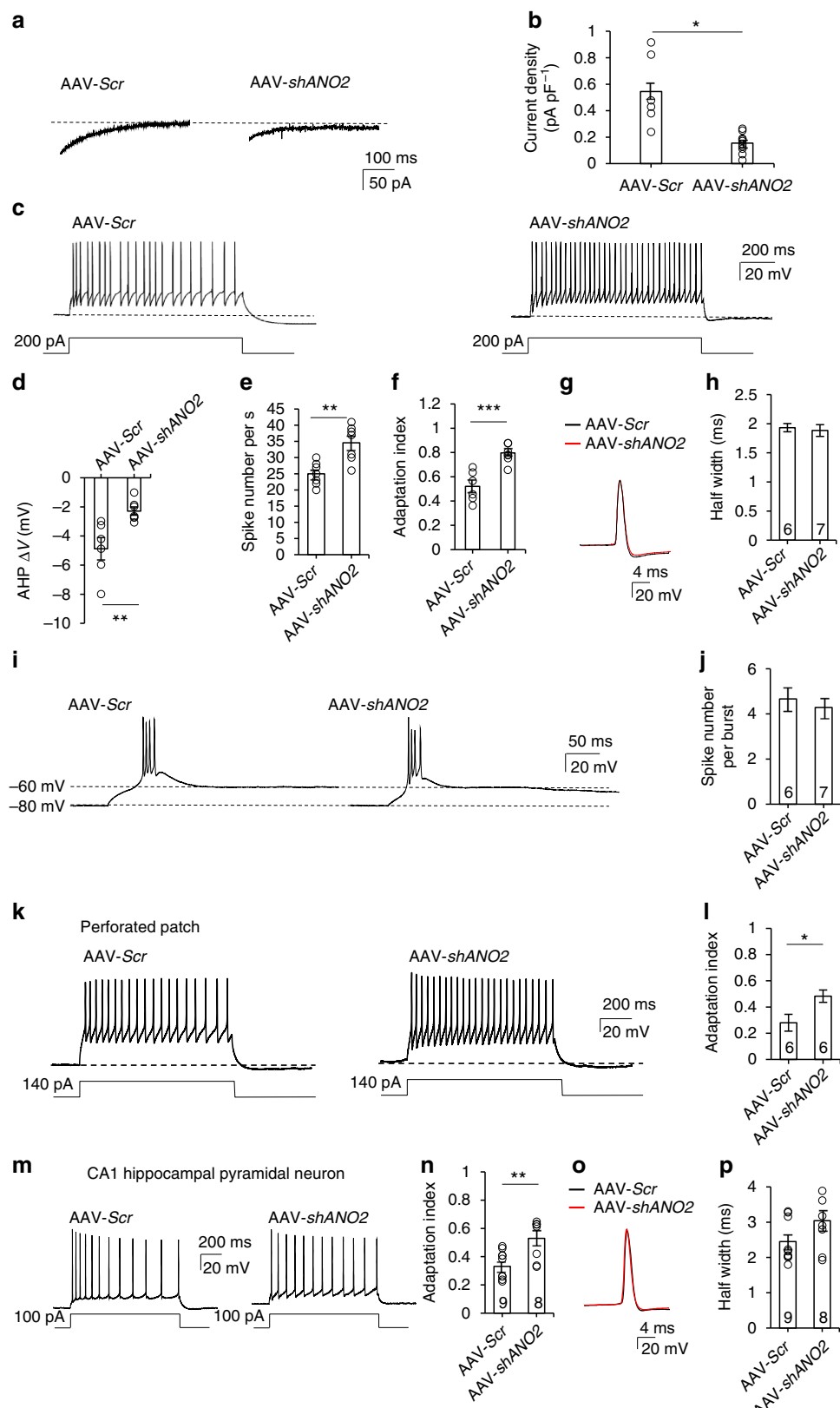

three mice, $P < 0.05$ by $t$-test, Fig. 7l). This result shows that ANO2 modulates spike-frequency adaptation under physiological condition. AAV-*shANO2*-infected CA1 hippocampal pyramidal neurons similarly showed a decrease in spike-frequency adaptation pattern (Fig. 7m) with an increased AI ($0.33 \pm 0.03$, $n = 9$ from three mice versus $0.53 \pm 0.05$, $n = 8$ from three mice, $P < 0.05$ by $t$-test, Fig. 7n) without altering the half-width of action potential ($2.45 \pm 0.20$ ms, $n = 9$ from three mice versus $3.04 \pm 0.28$ ms, $n = 8$ from three mice, $P = 0.11$ by $t$-test; Fig. 7o,p), suggesting that ANO2-mediated spike-frequency adaptation is a general mechanism.

Together, these results suggest that ANO2 mediation of $Ca^{2+}$-activated AHP currents suppresses the tonic-firing rate and produces a concomitant lengthening of ISIs, following a barrage of tonic spikes generated by highly activated TC neurons. On the basis of these observations, we further tested whether an increased tonic-firing rate affects sensory information transmission *in vivo*.

**Knockdown of ANO2 in VB nuclei increases pain responses.** To determine whether ANO2 currents regulate information transmission from the thalamus to the cortex *in vivo*, we examined visceral pain responses induced by intraperitoneal injections of acetic acid (0.6%) into mice with thalamic-restricted knockdown of ANO2. Pain behaviour tests were performed 3 weeks after injecting either AAV-*shANO2* or AAV-*Scr* bilaterally into the VB regions of the thalamus in 9–10-week-old mice (Fig. 8a). Only data obtained from mice with $70.2 \pm 3.9\%$ (minimum, 57.8%) knockdown efficacy in VB nuclei (confirmed by *post hoc* analysis of mCherry expression; Supplementary Fig. 7 and Supplementary Note 7) were included in behavioural analyses.

Writhing behaviours, such as stretching and constriction that have been described previously[16], were scored by investigators blinded to the treatment (Fig. 8b). Each writhing response was plotted as a bar, and the pattern showed that control mice (AAV-*Scr*) experienced pain-on periods interrupted by pain-off periods lasting tens of seconds; however, mice with thalamic-restricted knockdown of ANO2 (AAV-*shANO2*) experienced relatively short pain-off periods (Fig. 8c). Overall, ANO2 knockdown mice exhibited more persistent and frequent pain responses compared with control mice (AAV-*Scr* $n = 6$, AAV-*shANO2* $n = 8$; Fig. 8c). The number of writhing responses were significantly greater in AAV-*shANO2* mice than those in AAV-*Scr* mice (Fig. 8d), while there was no difference in the response between groups on the hot plate test (Supplementary Methods), which

measures acute pain response (Supplementary Fig. 8 and Supplementary Note 8). These results suggest that knockdown of ANO2 in TC neurons increases pain responses via diminished spike adaptation in TC neurons in persistent types of pain, such as visceral pain; however, the effect may be less evident in acute pain.

**Discussion**

We found that spike-frequency adaptation in TC neurons is mediated by the CACC ANO2. This is an unprecedented finding because it is the first report of a CACC-mediating neuronal spike-frequency adaptation in the central nervous system. The endogenous $Cl^-$ reversal (Fig. 4) results indicate that ANO2 mediates AHP current conduction in TC neurons, which might be assisted by the outwardly rectifying characteristic of ANO2 channels (Fig. 5e). The effects of CACCs have previously been ascribed to membrane depolarization via $Cl^-$ efflux in sensory neurons. Thalamic-restricted knockdown of ANO2 provided compelling evidence that ANO2 indeed mediates a gradual increase in ISIs (Fig. 7). Mice with VB-restricted ANO2 knockdown exhibited significant increases in pain responses, which strongly suggests that spike adaptation in TC neurons contributes to limiting excessive transmission of information by persistent activation. It is also noteworthy that our electrophysiological recordings were obtained in TC neurons from 5 to 7-week-old mice, for consistency with the behavioural data, whereas many previous studies recorded TC neurons from 2 to 3-week-old mice[3,37].

The mAHP current is characterized by its appearance following a lasting activation period with a duration ranging from tens to hundreds of milliseconds. These properties distinguish mAHP currents from both fast AHP currents, which appear at the end of each action potential, and slow AHP currents, which last for seconds[38]. In addition, mAHP currents have typically been ascribed to $Ca^{2+}$-activated $K^+$ channels[39]. SK channels in the hippocampus[5] and TRN[24] conduct $Ca^{2+}$-dependent AHP currents via interactions with specific voltage-gated $Ca^{2+}$ channels or via intracellular $Ca^{2+}$ release. Once SK channels in the TRN are activated by T-type $Ca^{2+}$ currents, they generate robust hyperpolarization lasting tens of milliseconds, followed by re-activation of T-type channels[24]. This interaction between T-type and SK channels generates recurrent oscillatory activity in TRN neurons[24]. Our data also demonstrate that the AHP component in TC neurons is partially mediated by SK channels. Blocking SK channels decreased mAHP currents and increased the tonic spike rate, but had limited influence on spike

**Figure 7 | Effect of TC-specific knockdown of ANO2 on spike adaptation and mAHP currents in TC neurons.** (**a**) AHP currents were measured in VB TC neurons infected with *Scr* or *shANO2* AAV viruses. (**b**) *shANO2*-infected neurons ($n = 10$ from three mice) displayed significantly lower current densities compared with AAV-*Scr*-infected neurons ($n = 6$ from six mice; *$P = 0.014$, $t$-test). (**c**) Tonic firing and AHP were generated by current injection in TC neurons infected with AAV-*Scr* or AAV-*shANO2*. (**d**) AHP potentials in AAV-*Scr*- ($n = 6$ from five mice) or AAV-*shANO2*-infected ($n = 7$ from five mice) TC neurons were quantified (**$P = 0.009$, $t$-test). (**e,f**) In *shANO2*-infected TC neurons, the AHP after current injection decreased, whereas the number of spikes (**$P = 0.009$, $t$-test) and the adaptation indexes increased (***$P = 0.0008$, $t$-test). (**g**) Single action potentials from AAV-*Scr* (black) and AAV-*shANO2* (red) infected TC neurons. (**h**) There was no difference in the half width of single action potentials between AAV-*Scr*-infected (black, $n = 6$ from five mice) and AAV-*shANO2*-infected (red, $n = 7$ from five mice) TC neurons ($P = 0.46$, $t$-test). (**i**) Representative traces of burst firing on resting membrane potential ($\sim -60$ mV) released from hyperpolarization ($\sim -80$ mV). (**j**) Patterns and number of spikes per burst were similar in AAV-*Scr*-infected ($n = 6$ from five mice) and AAV-*shANO2*-infected ($n = 7$ from five mice) TC neurons ($P = 0.57$, $t$-test). (**k**) Under intact $[Cl^-]_{in}$ conditions performed by perforated patch, tonic firing in AAV-*Scr* and AAV-*shANO2* TC neurons were generated by 140 pA current injection. (**l**) An analysis of adaptation index in both AAV-*Scr*-infected ($n = 6$ from three mice) and AAV-*shANO2*-infected ($n = 6$ from four mice) TC neurons showed that AAV-*shANO2* infection induced a significant decrease in spike-frequency adaptation (*$P = 0.028$, $t$-test). (**m**) Tonic firing was generated by current injection in CA1 hippocampal pyramidal neurons infected with AAV-*Scr* or AAV-*shANO2*. (**n**) Adaptation indexes were averaged in both AAV-*Scr*-infected ($n = 9$ from three mice) and AAV-*shANO2*-infected ($n = 8$ from three mice) neurons (**$P = 0.0098$, $t$-test). (**o**) Single action potentials from AAV-*Scr* (black) and AAV-*shANO2* (red) infected CA1 neurons. (**p**) There was no difference in the half-width of single action potentials between AAV-*Scr*-infected (black, $n = 9$ from three mice) and AAV-*shANO2*-infected (red, $n = 8$ from three mice) CA1 neurons ($P = 0.11$, $t$-test). Error bars represent mean ± s.e.

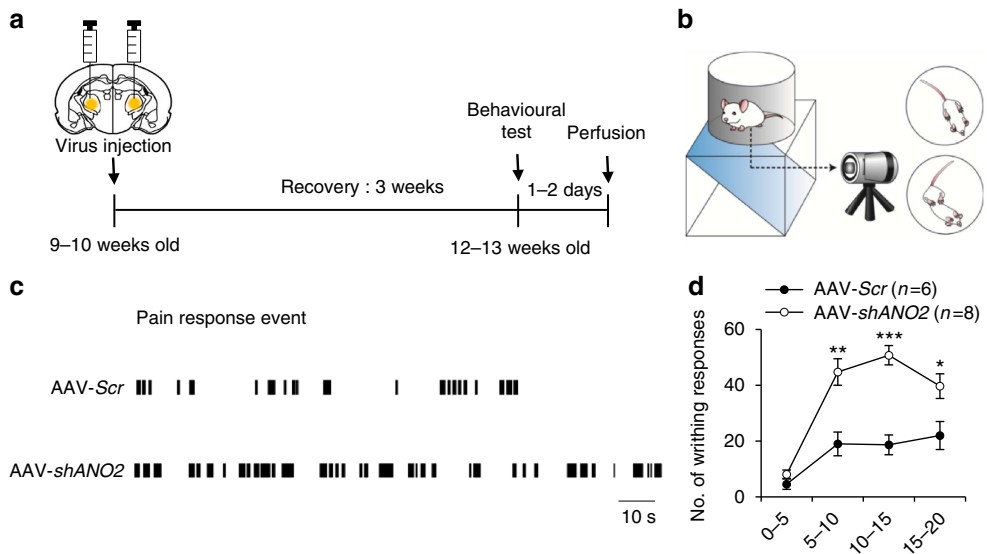

**Figure 8 | Modulation of visceral pain responses by expression of ANO2. (a)** Timeline for behavioural testing demonstrating that tests were conducted 3 weeks after viral injection into TC neurons. Mouse brains were isolated to confirm the injection site and the efficiency of knockdown 1–2 days after behavioural testing. **(b)** Schematic image of the visceral pain response paradigm. **(c)** Representative data for the writhing response over a 10–15 min period demonstrated that AAV-shANO2 mice displayed longer and more frequent pain responses than AAV-Scr mice. **(d)** The number of writhing responses in mice injected with AAV-Scr ($n = 6$) or AAV-shANO2 ($n = 8$) was scored in 5-min intervals. VB-specific knockdown of ANO2 induced an increase in visceral pain responses in vivo (*$P = 0.037$, **$P = 0.009$, ***$P = 0.0001$, $t$-test). Error bars represent mean ± s.e.

adaptation in TC neurons. A relatively small contribution of SK channel-mediated mAHP currents in TC neurons also explains why recurrent oscillatory burst firing is not observed in TC neurons. In contrast, thalamic-specific knockdown of ANO2 significantly reduced spike adaptation, clearly indicating that mAHP currents in TC neurons are highly regulated by ANO2 (Fig. 7), although we could not completely exclude the possibility that other $K^+$ channels could also contribute to AHP[26,40]. This notion was also confirmed in CA1 hippocampal neurons (Fig.7). The functional differences between SK and ANO2 in TC neurons may result from differences in decay kinetics. SK currents in TC neurons generate an earlier peak and decay faster than ANO2 currents (Fig. 2a), indicating that ANO2 more effectively suppresses spike adaptation because of its longer duration. The distinctive functions of the two channel types may also result from differences in their localization. SK channels are expressed in dendrites and, therefore, may reduce the excitability of TC neurons by hyperpolarizing the membrane of dendrites without mediating spike adaptation; this notion is supported by our data (Fig. 3). In contrast, ANO2 appears to be localized to the soma of TC neurons, suggesting a contribution to spike-frequency tuning.

ANO2, in olfactory epithelium[32] and retinal photoreceptors[33], mediates transepithelial $Cl^-$ secretion and presynaptic $Ca^{2+}$-activated depolarization of the membrane potential via efflux of $Cl^-$. In contrast with these previous studies, the present study provides evidence that $Ca^{2+}$-activated $Cl^-$ conductance via ANO2 channels hyperpolarizes the membrane potential. Multiple factors appear to be involved in this process. First, the reversal potential of $Cl^-$ in TC neurons with intact intracellular $Cl^-$ content was about $-70$ mV. This potential approximates the resting membrane potential of TC neurons and is also lower than the threshold required to activate voltage-gated $Na^+$ channels. The $E_{Cl}$ indicated low $[Cl^-]_{in}$ in TC neurons, which is consistent with previous studies demonstrating that the $K^+/Cl^-$ co-transporter (KCC2) is highly expressed in TC neurons and actively results in $Cl^-$ efflux[41] leading to

hyperpolarization of $E_{GABA}$ (ref. 37). Second, ANO2 generates an outwardly rectifying current (Fig. 5e), while ANO1 manifests a linear $I$–$V$ curve[42]. Therefore, ANO2 would generate relatively large outward currents in TC neurons with an endogenous ionic content at the depolarized membrane, although in our study we intentionally measured a small inward current using a high $Cl^-$ intrapipette solution in order to easily distinguish $Cl^-$ components from $K^+$ components in $Ca^{2+}$-activated tail currents. Third, ANO2 activation is relatively less sensitive to $Ca^{2+}$ compared with ANO1 (ref. 43). These data indicate a unique role for ANO2 in controlling the excitability of TC neurons. When TC neurons generate spikes at a low frequency, the membrane potential of TC neurons remains close to the reversal potential of $Cl^-$ with low levels of $Ca^{2+}$ influx. Therefore, ANO2 does not conduct a substantial current, and TC neurons generate spikes at regular intervals. When TC neurons generate a barrage of spikes at high frequencies, the membrane of the TC neurons is depolarized, which is accompanied by high levels of $Ca^{2+}$ influx near the soma. Hyperpolarization of the membrane potential then occurs via the ANO2 current, which elongates the ISIs. Therefore, ANO2 currents restrict excessive spike generation but do not interfere with information transmission by TC neurons up to a certain level of spike frequency.

Our experiments revealed that spike adaptation and mAHP currents induced by strong depolarizing inputs to the TC neurons are almost completely abolished by replacing the extracellular buffer with $Ca^{2+}$-free buffer (Figs 1 and 2), indicating that ANO2 interacts with voltage-gated $Ca^{2+}$ channels. Our data also demonstrate that $Ca^{2+}$-activated tail currents are maximal under 0 mV steps, suggesting an involvement of high-voltage-gated $Ca^{2+}$ channels. However, there was no change in the low-threshold spikes of TC neurons following ANO2 knockdown, indicating that T-type channels do not contribute to generating these tail currents. This finding may be explained further by the primary expression of ANO2 in the soma of TC neurons (Fig. 6c), while T-type channels are expressed at

approximately uniform density throughout the soma and dendrites of TC neurons[44,45]. Another potential explanation is that small and transient $Ca^{2+}$ influx through T-type $Ca^{2+}$ channels near the soma is insufficient to activate ANO2, which has relatively low $Ca^{2+}$ sensitivity. A previous report indicates that blocking L-type channels, the most abundant high-voltage-gated $Ca^{2+}$ channels in TC neurons, increased tonic-firing rates and pain responses[25]; this concurs with the present data.

The role of ANO2-mediated spike adaptation, which can be considered a type of self-inhibition in TC neurons, was emphasized on the basis of a considerable increase in pain responses in mice with thalamic-restricted ANO2 knockdown (Fig. 8). Visceral pain, relayed through VB nuclei in the thalamus to the somatosensory cortex[46], has previously been shown to be modulated by the alteration of thalamic spiking patterns in rodents[2,16]. Once pain responses began, animals in the control group experienced repeated pain-on and pain-off periods (Fig. 8c). Prolonged durations of pain-on experiences in ANO2 knockdown mice support the hypothesis that spike adaption contributes to suppressing excessive transmission of information from the thalamus to the cortex. Spike adaption in neurons has been suggested as a crucial contributor to stimulus encoding by neurons[47,48]. Specifically, spike adaptation may enable neurons to respond more sensitively to coinciding inputs[49] or have a major contribution to network synchronization[50], which suggests that ANO2-mediated spike-frequency adaptation in TC neurons may facilitate synchronized TC activity.

TC neurons project sensory information to cortical neurons; therefore, alterations in tonic firing from regular spiking to spike adaptation likely contribute to various types of sensory information processing, including pain transmission. This change may contribute to desensitization of sensory responses that occur with long-lasting or repetitive sensory stimuli in various sensory modalities. The reversal of ANO2 currents that are close to the resting membrane potential may have resulted in a previous underestimation of the role of $Cl^-$ currents in regulating neuronal excitability. Additional studies may reveal novel functions of CACCs in TC neurons and provide additional information on the nature of sensory information processing in TC circuits.

## Methods

**Animals.** For patch clamp and molecular biology experiments, both male and female BALB/c mice (4–6 weeks old) were used. Eight- to nine-week-old male mice were used for behavioural test. Best1 knockout ($BEST1^{-/-}$) mice and wild-type littermates were obtained by mating heterozygous mice with BALB/c genetic backgrounds. Genotypes were determined by PCR analyses using the following primers: mb0601r, 5′-TGAATGGTGACCTCCAAGGCTATCC-3′; mb9374f, 5′-AACAAAGGGTAAGCAGGAGTGCCC-3′; and neo2351f, 5′-GGAAGACAA TAGCAGGCATGCTGGG-3′. GFAP-GFP (green fluorescent protein) transgenic mice were kept on FVB genetic backgrounds. Genotypes were determined by PCR analyses using the following primers: 42, 5′-CTAGGCCACAGAATTGAAAGA TCT-3′; 43, 5′-GTAGGTGGGAAATTCTAGCATCATCC-3′; 872, 5′-AAGTTCATC TGCACCACCG-3′ and 1416, 5′-TCCTTGAAGAAGATGGTGCG-3′. All mice were maintained under a 12:12-h light–dark cycle (lights on 7:00 A.M.) and had *ad libitum* access to food and water. Care and handling of animals was in accordance with the guidelines of the Institutional Animal Care and Use Committee at Yonsei University (Seoul, Korea).

**Preparation of brain slices.** Firing and AHP currents of TC neurons were recorded in acutely isolated horizontal brain slices from 4- to 6-week-old mice. Mice were anaesthetized with halothane and decapitated to remove the brain. The brains were sectioned in ice-cold slicing solution (sucrose, 234 mM; KCl, 2.5 mM; MgSO$_4$, 10 mM; NaH$_2$PO$_4$, 1.25 mM; NaHCO$_3$, 24 mM; CaCl$_2$·2H$_2$O, 0.5 mM; and glucose, 11 mM). Horizontal slices (300 µm thick) were sectioned with a vibrating-knife microtome VT1000s (Leica Microsystems, Germany). For stabilization, slices were incubated for at least 1 h in a solution containing NaCl, 124 mM; KCl, 3 mM; MgSO$_4$, 6.5 mM; NaH$_2$PO$_4$, 1.25 mM; NaHCO$_3$, 26 mM; CaCl$_2$·2H$_2$O, 1 mM; and glucose, 10 mM, and simultaneously equilibrated with 95% O$_2$/5% CO$_2$

at 25 °C. Most of the electrophysiological recordings from the brain slices were performed at 25 °C except Supplementary Fig. 1c,d, which was performed at 32 °C.

**Cell visualization and data acquisition.** Individual cells were visually identified using an upright Olympus EX51WI (Olympus, Japan) microscope equipped with an ORCA-R2 camera (HAMAMATSU, Japan). Signals were amplified using MultiClamp 700B (Molecular Devices, USA), and data acquisition was performed using a Digitizer 1440A and Clampex (Molecular Devices). All data analyses were performed using the Clampfit (Molecular Devices) and Mini Analysis software (Synaptosoft Inc., USA).

**Induction of firing and AHP.** Firing of VB thalamic neurons in brain slices was recorded under current clamp in aCSF solution (NaCl, 124 mM; KCl, 3 mM; MgSO$_4$, 1.3 mM; NaH$_2$PO$_4$, 1.25 mM; NaHCO$_3$, 26 mM; CaCl$_2$·2H$_2$O, 2.4 mM; and glucose, 10 mM) aerated with 95% O$_2$/5% CO$_2$. Patch electrodes (4–6 MΩ) fabricated from standard-wall borosilicate glass (GC150F-10, Warner Instrument Corp., USA) were filled with an intrapipette solution containing K-gluconate, 125 mM; KCl, 10 mM; MgCl$_2$, 1 mM; HEPES, 10 mM; EGTA, 0.02 mM; Mg-ATP, 4 mM; and Na$_2$-GTP, 0.3 mM, with pH adjusted to 7.35.

**$Ca^{2+}$ dependency of AHP currents.** $Ca^{2+}$-dependent membrane potentials of AHP and $Ca^{2+}$-dependent AHP currents were recorded using a whole-cell patch clamp configuration from acute brain slices in aCSF solution containing NaCl, 124 mM; KCl, 3 mM; MgSO$_4$, 1.3 mM; NaH$_2$PO$_4$, 1.25 mM; NaHCO$_3$, 26 mM; CaCl$_2$·2H$_2$O, 2.4 mM; and glucose, 10 mM. Under voltage clamp, the holding potential was −60 mV, and voltage steps ranging from −60 to +40 mV (+10 mV per step) were applied every 50, 100, 200, 500 and 1,000 ms. This process occurred in both 2.4 mM $Ca^{2+}$ buffer NaCl, 124 mM; KCl, 3 mM; MgSO$_4$, 1.3 mM; NaH$_2$PO$_4$, 1.25 mM; NaHCO$_3$, 26 mM; CaCl$_2$·2H$_2$O, 2.4 mM; and glucose, 10 mM) and $Ca^{2+}$-free buffer (NaCl, 124 mM; KCl, 3 mM; MgSO$_4$, 1.3 mM; NaH$_2$PO$_4$, 1.25 mM; NaHCO$_3$, 26 mM; EGTA, 5 mM; and glucose, 10 mM). The intrapipette solution contained KCl, 60 mM; K-gluconate, 73 mM; MgSO$_4$, 1 mM; HEPES, 10 mM; EGTA, 0.1 mM; Mg-ATP, 4 mM; and Na$_2$-GTP, 0.3 mM (pH 7.35 and 285 mOsm).

**Pharmacological dissection of AHP currents.** For dissection of AHP currents, the intrapipette solution contained KCl, 60 mM; K-gluconate, 73 mM; MgSO$_4$, 1 mM; HEPES, 10 mM; EGTA, 0.1 mM; Mg-ATP, 4 mM; and Na$_2$-GTP, 0.3 mM (pH 7.35 and 285 mOsm). TTX (0.5 µM), apamin (0.1 µM) and/or NFA (100 µM) were applied after achieving the whole-cell configuration. The bath solution was treated with TTX for recordings of $Ca^{2+}$-dependent AHP currents and was perfused with TTX and apamin in order to obtain apamin-sensitive $K^+$ currents. For dissection of NFA-sensitive anion currents, recording aCSF was perfused with TTX, apamin and NFA. Apamin-sensitive $K^+$ currents and NFA-sensitive anion currents were calculated by trace subtraction.

**Reversal potential of AHP currents.** Reversal potentials for AHP currents were measured with intrapipette solutions containing various levels of [Cl$^-$]$_{in}$ ranging from 12 to 60 mM in voltage clamp. The high [Cl$^-$]$_{in}$ condition contained KCl, 60 mM; K-gluconate, 73 mM; MgSO$_4$, 1 mM; HEPES, 10 mM; EGTA, 0.1 mM; Mg-ATP, 4 mM; and Na$_2$-GTP 0.3 mM (pH 7.35 and 285 mOsm). The moderate [Cl$^-$]$_{in}$ condition contained K-gluconate, 115 mM; KCl, 20 mM; MgCl$_2$, 1 mM; HEPES, 10 mM; EGTA, 0.02 mM; Mg-ATP, 4 mM; and Na$_2$-GTP 0.3 mM (pH 7.35 and 280–290 mOsm). The low [Cl$^-$]$_{in}$ condition contained K-gluconate, 125 mM; KCl, 10 mM; MgCl$_2$, 1 mM; HEPES, 10 mM; EGTA, 0.02 mM; Mg-ATP, 4 mM; and Na$_2$-GTP 0.3 mM (pH 7.35 and 281 mOsm). The junction potentials were measured and corrected for all the intrapipette solution and extracellular buffer pairs used for patch clamp recordings.

**Perforated patch clamping.** For perforated patch clamp recordings, the pipette solution contained gramicidin-D, 50 µg ml$^{-1}$; KCl, 140 mM; HEPES, 10 mM; Mg-ATP, 4 mM; and Na$_3$-GTP, 0.3 mM (pH 7.2 and 304 mOsm). The extracellular solution consisted of NaCl, 130 mM; NaHCO$_3$, 24 mM; KCl, 3.5 mM; NaH$_2$PO$_4$, 1.25 mM; CaCl$_2$, 1 mM; MgCl$_2$, 3 mM; and glucose, 10 mM (pH 7.4 and 320–330 mOsm). It took ∼20–30 min to achieve acceptable perforation, with final series resistances ranging from 15 to 30 MΩ. The membrane holding potential was −60 mV unless otherwise specified.

**MONNA selectivity test.** HEK293T cells (human embryonic kidney 293T cells, ATCC, USA) were tested for mycoplasma and culture in DMEM (Gibco, USA), containing 10% fetal bovine serum (Gibco), penicillin (100 U ml$^{-1}$, Gibco) and streptomycin (100 µg ml$^{-1}$, Gibco). They were transfected using the Effectene Transfection Reagent (Qiagen, Germany) containing 400 ng of mANO1-IRES-GFP or mANO2-EGFP-N1 for 16 h. On the day of the experiment, the transfected cells were trypsinized and transferred on poly-D-lysine-coated coverslips in a 12-well plate. The coverslip with attached cells was placed on the stage of a BX51WI Olympus microscope (Japan) and perfused with HEPES-based solution. The

extracellular solution was composed of HEPES, 10 mM; NaCl, 150 mM; KCl, 3.2 mM; CaCl$_2$, 2 mM; MgCl$_2$, 2 mM; glucose, 5.6 mM; and sucrose, 22 mM (pH adjusted to 7.2 with NaOH). Visually guided whole-cell patch recordings were obtained from GFP-positive HEK293T cells using borosilicate patch pipettes filled with high Ca$^{2+}$ internal solution CsCl, 146 mM; Ca-EGTA-NMDG, 5 mM; MgCl$_2$, 2 mM; HEPES, 8 mM; sucrose, 10 mM; Na-ATP, 4 mM; and Na-GTP, 0.3 mM (pH adjusted to 7.3 with CsOH). Data were collected with a Multiclamp 700B amplifier (Molecular Devices) using the Clampex 9 acquisition software and digitized with Digidata 1322A (Molecular Devices).

**Data analysis.** Neurons with series resistance < 30 MΩ were accepted for statistical analysis. The normality test was run before data analysis. The membrane properties of recorded neurons were summarized in the Supplementary Table 1. Most of the data met the normality criteria and the data are presented as mean ± s.e.m. Two-tailed $t$-tests were used for statistical analyses, unless otherwise specified. Firing patterns of TC neurons and Ca$^{2+}$ current data were analysed using the Mini Analysis software (Synaptosoft Inc.).

**shRNA design and vector construction.** A mouse ANO2 sequence (NM_153589) from 1,137 to 1,157 (5′-GCCGCATTGTACACGAGATTC-3′) was selected as the target region of shRNA. The pENSR-shANO2-mCherry construct was synthesized using complementary oligomers: 5′-agagaGAATCTCGTGTACAATGCGGCtttttttc tcgagtactagga-3′ (sense) and 5′-tgaaGAATCTCGTGTACAATGCGGCaaaca aggcttttctccaag-3′ (antisense). Oligomers were ligated with the pENSR backbone using site-directed mutagenesis (Enzynomics, Korea) and verified by sequencing. Scrambled shRNA-containing constructs were used as controls: 5′-ttcgcatagcgta tgccgttttcaagagaaacggcatacgctatgcgattttttc-3′ (sense) and 5′-tcgagaaaaaatcgcatag cgtatgccgtttctcttgaaaacggcatacgctatgcgaa-3′ (antisense).

**Virus infection.** Approximately 3–4-week-old mice were used for patch clamp experiments and 8–9-week-old mice were used for *in vivo* experiments. Mice were anaesthetized via intraperitoneal (i.p.) injections of 2,2,2-tribromoethanol (300 mg kg$^{-1}$ body weight; Sigma-Aldrich, USA). AAV-shANO2-mCherry or AAV-Scr-mCherry was bilaterally microinjected into the VB region of the thalamus (for ~ 3–4-week-old mice, − 1.3 mm lambda from bregma, ± 1.4 mm lateral to the midline, − 3.4 and − 2.9 mm ventral from the brain surface; for adult mice, − 1.8 mm lambda from bregma, ± 1.5 mm lateral to the midline, − 3.5 and − 2.5 mm ventral from the brain surface). Mice were used for experiments 2 weeks following injections.

**Transfection.** As a positive control for both RT–PCR and western blotting experiments, NIH/3T3 cells (mouse fibroblast cells, ATCC) were transfected with a plasmid expressing mouse ANO2-EGFP using Lipofectamine 2000 (Invitrogen, USA). NIH/3T3 cells tested for mycoplasma were cultured in DMEM (Gibco), containing 10% bovine serum (Gibco), penicillin (100 U ml$^{-1}$, Gibco) and strep-tomycin (100 μg ml$^{-1}$, Gibco). When NIH/3T3 cells were 70–80% confluent, they were incubated in a mixture of ANO2-EGFP vector with Lipofectamine 2000 for 24 h before the media was changed. Cells were harvested 2 days after transfection for use in RT–PCR and western blotting experiments.

**RT-PCR.** Total RNA was isolated from the olfactory epithelium and thalamus of BALB/c mice, as well as from ANO2-EGFP-transfected and untransfected NIH/3T3 cells, using an RNA isolation kit (Hybrid-R, GeneAll, Korea). One microgram of total RNA was used for cDNA synthesis with the SuperScript III First-Strand Synthesis System for RT–PCR (Invitrogen). PCR primers used to confirm expression were as follows: β-actin; F1 primer 5′-TGTGATGGTGGGAATGGGT CAGAA-3′ and R1 primer 5′-TGTGGTGCCAGATCTTCTCCATGT-3′ to produce a 140 bp cDNA; ANO2, F2 primer 5′-AGAGCCTTACACTGGGCTGA-3′ and R2 primer 5′-GTCCCGTTGTGGCTATAGGA-3′ to produce a 321 bp cDNA. PCR conditions were as follows: 5 min at 94 °C for one cycle, and then 35 cycles of 30 s at 94 °C, 30 s at 60 °C, 1 min at 72 °C, followed by 5 min at 72 °C.

**Western blot analysis.** The thalamus and kidney from BALB/c mice, as well as ANO2-EGFP-transfected and untransfected NIH/3T3 cells, were lysed in RIPA buffer (Tris-base, 6.5 mM; NaCl, 15 mM; EDTA, 1 mM; NP-40, 1%; and Na-deoxycholate, 0.25%) with 1 mM phenylmethyl sulphonyl fluoride, Na$_3$VO$_4$, NaF and protease cocktail. Lysates were centrifuged at 13,000 r.p.m. for 10 min at 4 °C, and the supernatants were isolated for SDS–PAGE. From each sample, 50 μg of protein was resolved on 8% polyacrylamide gels and subsequently electroblotted to polyvinylidene fluoride (Bio-Rad, USA) membranes. Blots were incubated with primary antibodies overnight at 4 °C, followed by incubation with horseradish peroxidase (HRP)-linked secondary antibodies. Signals were visualized using ECL Plus reagent (Thermo Scientific, USA). The following antibodies were used: rabbit anti-ANO1 (ab53212, 1:500, Abcam, UK), rabbit anti-ANO2 (1:500 (ref. 35)), mouse anti-β-actin (sc-47778, 1:2,000, Santa Cruz, USA), goat anti-rabbit-HRP-conjugated (31460, 1:5,000, Pierce, USA) and goat anti-mouse-HRP-conjugated (31430, 1:5,000, Pierce).

**Immunohistochemistry.** Mice were anaesthetized via i.p. injection of 2,2,2-tri-bromoethanol (300 mg kg$^{-1}$, Sigma-Aldrich) and transcardially perfused with 1 M PBS followed by a 4% paraformaldehyde (PFA) solution. Brains were isolated from mice and post-fixed in a 4% PFA solution overnight, followed by immersion in a 30% sucrose solution for 48 h to achieve cryoprotection. Brains mounted in OCT compound (Sakura, USA) were sliced coronally with a cryostat to obtain brain sections (40 μm) containing the ventral posterolateral nucleus/ventral poster-omedial nucleus thalamic region.

The sections were incubated in 0.1% Tween-20 in 1 M PBS for 30 min and then in blocking solution (5% normal goat serum in 1 M PBS) for 1 h, followed by primary antibodies with blocking solution overnight at 4 °C. After incubation with primary antibodies, sections were incubated in secondary antibody with blocking solution for 2 h at 25 °C before being mounted on microscope slides with Fluorescence Mounting Medium (DAKO, USA). The following antibodies were used: rabbit anti-BEST1 (1:100; ref. 51), rabbit anti-ANO1 (ab64085, 1:300; Abcam), rabbit anti-ANO2 (1:500; ref. 35), mouse anti-NeuN (MAB377, 1:400; Millipore, Germany), chicken anti-GFP (ab13970, 1:1,000; Abcam), cy3-labelled goat anti-rabbit IgG (A10520, 1:500; Life Technologies, USA), DyLight 405 goat anti-mouse IgG (115-475-003, 1:500; Jackson ImmunoResearch, USA), and Alexa 488 goat anti-chicken IgY (A11039, 1:500; Life Technologies). Fluorescence images were obtained with an LSM 700 confocal microscope (Carl Zeiss, Germany).

**Image quantification.** Confocal microscopic images were obtained in order to quantify reductions in ANO2 expression and were analysed using the ImageJ (NIH) programme. ANO2 expression in virus-infected TC neurons was calculated using the mean intensity value of ANO2-immunostained pixels in the mCherry-positive area. The mCherry-positive area was defined by thresholding and is converted into a binary mask. The mean intensity of ANO2-immunostained pixels in the binary mask was calculated.

**Visceral pain test.** Visceral pain was induced by i.p. injection of 6 ml kg$^{-1}$ acetic acid (0.6% in saline). Experimental groups were not disclosed to either the investigator who performed the pain tests or the evaluators who scored the responses in order to prevent bias. Pain responses were recorded in a random order with a charge-coupled device camera, and the results were analysed later by two blinded evaluators. Experimental groups were matched with the results of the pain assay after evaluations were complete. The mice whose AAV-infected VB region is below 50% were excluded for behavioural analysis.

**Data availability.** The data that support the findings of this study are available from the authors upon reasonable request.

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

## Acknowledgements

We appreciate T.J. Jentsch and J. Muench for kindly providing antibodies and valuable discussions. We are grateful to our colleague, Soojung Lee, for helpful technical support and valuable discussions. This research was supported by the National Research Foundation (NRF-2014R1A2A2A01006940, NRF2015R1A3A2066619 and NRF-2014M3A7 B4051596), funded by the government of the Republic of Korea (Ministry of Science, ICT and Future Planning, MSIP), International Collaborative R&D Program, funded by the Ministry of Trade, Industry and Energy (MOTIE, Korea), the Yonsei University Future-Leading Research Initiative of 2015 (2015-22-0163), KIST Institutional Program (2E26664), Samsung Research Funding Center of Samsung Electronics under Project Number SRFC-IT1402-08 and the Brain Korea 21 (BK21) PLUS program. G.E.H., H.K., G.-E.C. and J.H. are fellowship awardee by the BK21 PLUS program.

## Author contributions

E.C. conceived the project. G.E.H., J.L., H.K., K.S., J.K., S.-Y.J., J.H., G.-E.C. and E.M.H. performed the experiments and analysed the data. G.E.H. managed the direction of the project and contributed to immunohistochemistry, western blotting, RT–PCR and pain response analyses. E.M.H. contributed to immunohistochemistry and western blotting. J.L., H.K., K.S. and G.-E.C. contributed to the electrophysiology experiments, including whole-cell patch clamping. J.L. performed the perforated patch experiments. J.K. contributed to the development and screening of shRNA and scrambled RNA. S.-Y.J. contributed to the MONNA selectivity test, and G.E.H. and J.H. performed viral injections *in vivo*. H.-S.S., C.J.L. and E.C. supervised the experimental analyses and manuscript preparation.

## Additional information

**Competing financial interests:** The authors declare no competing financial interests.

