## [Peer Review File · Nature Communications]

Reviewers' comments:

Reviewer #1 (Remarks to the Author):

This is an interesting and potentially important study in its field. The authors show that spike-frequency adaptation in thalamocortical neurons is mediated by the Ca²⁺-activated Cl⁻ channel (CACC) anoctamin-2 (ANO2). This finding is (to the best knowledge of the present referee) a novel one. However, the paper has shortcomings which should be carefully addressed by the authors. Most of these can be overcome by text revisions, but perhaps one experiment could be added as explained in the first point in the list below.

Major comments

- 1) Given the key role of intracellular Cl (Cl_i), it would be interesting if the authors would provide data on spike-frequency adaptation (SFA) under conditions where Cl_i would be intact and prone to activity-dependent variation. The gramicidin patch technique has been used only in experiments using GABA to probe the steady-state level of Cl_i and the resulting E_{Cl}. For studying SFA under conditions with intact Cl_i, on-cell patching might also be an effective approach. Anyway, it is entirely possible that even under in vitro conditions, somatic Cl_i would show activity-dependent (spike-firing dependent) fluctuations and thereby modulate both the ANO2-mediated current and, consequently, SFA. I do not insist on this experiment, but this is a point that should at least be taken up in the Discussion.
- 2) Surprisingly, the authors do not clearly state the major, general novelty of their work in the Abstract. On p. 9 they state that "no previous study has described a CACC that is involved in mAHP currents and neuronal spike patterns in the central nervous system". This important point should be included into the Abstract.
- 3) The adaptation index used presently will be extremely sensitive to minor changes in the first and last interspike interval. I would suggest using an average of at least two if not three ISIs (i.e., the initial 2 or 3 and the last 2 or 3 ISIs) two make the parameter more robust.
- 4) The authors should explain why pain responses only have been studied (section starting on p. 14). In thalamocortical sensory functions, other sensory modalities take a much more central place in the literature.
- 5) Discussion, start of: the data in Fig. 2d do not indicate that this is an outwardly rectifying current as stated in the text.
- 6) Ca²⁺ buffering in the pipette solution has been done with very low and variable amounts of EGTA. Because somatic Ca²⁺ is a key player, the authors should provide information on the possible effects (or attempts to minimize them) of the exogenous buffers. This would be relevant for the discussion on p. 19, 2nd paragraph.
- 7) p. 18: Outward rectification has nothing to do with a channel's capability to hyperpolarize the plasma membrane. While this kind of a misconception is frequent, it is the reversal potential only which dictates the polarity of a channel-mediated response.
- 8) There's a major confusion with regard to Fig. 7 where, for instance, panels j and k are not even mentioned in the Fig. legend. Please also cross-check with main text (p. 13).
- 9) The style of writing is verbose and sometimes the flow is somewhat clumsy. I would urge the authors to aim at a much more compact style of writing throughout the ms (especially the Results section). This would also make the paper shorter.

Further comments

- a) The title should state which cells are being studied
- b) The running title is not well-formulated. I would suggest "ANO2 and spike frequency adaptation" or something along these lines ("dampening information transfer" is vague/ambiguous)
- c) p. 4: "is still debated". Here the authors cite papers from 2001-2003. Novel references are needed to justify "still debated".
- d) p. 6, line from below: "holding current" should be "holding potential"
- e) p. 6, last line: these were surely not "0 mV prepulse steps" but rather "prepulse steps to 0 mV"

- f) p. 7 line 2: What are the error bars in the text and Figures? SD or SEM? I am not sure whether this information has been provided. Anyway, the numerical value 32 ± 7.1 ms does not match with what is shown in Fig. 1j. Perhaps similar errors are present elsewhere - please check.
- g) It is confusing to separate the upper panels of Fig. 1 into a and b, they are from the same cell.
- h) The voltage protocol used should be illustrated in Fig. 2c (which also lacks calibration ms/pA), and in Fig. 7j.
- i) p. 9: "These results indicate that SK channels contribute substantially to the generation of Ca²⁺-activated mAHPs, but are not critical to spike frequency adaptation." This sentence should be clarified (see capital letters below), perhaps like this: "These results indicate that SK channels contribute substantially to the generation of Ca²⁺-activated mAHPs AND TONIC SPIKE FIRING RATE, but are not.."
- j) p. 10, line 6: "...the intercept on..". Please state intercept with what - i.e. intercept with the voltage axis.
- k) The estimated 8.4 mM Cl_i has not been corrected for bicarbonate permeability. With an internal HCO₃ of 15 mM typical for CNS neurons, and a relative permeability of 0.2 or 0.3 vs Cl, the real Cl_i based on the Egaba measurements is around 5-6 mM. The authors might also note that ANO2 is a very unspecific anion channel which may carry substantial (depolarizing) HCO₃ current components (see Farrant and Kaila 2007 review for a quantitative treatment).
- l) p. 12, line 5: "remaining" should be "main".

Minor comments

- Abstract line 7: "by an increased tonic spikes" - please correct.
- p. 4, line 10 from below: "mAHP is are known" - please correct
- p. 4, line 5 from below: to make it easier for the reader, please replace "Ca²⁺ activating mAHP currents" by "Ca²⁺ currents which activate mAHP currents"
- p. 7, line 3: "elicited larger Ca²⁺ influxes" is not based on data; please reformulate.
- p. 8, line 3 from below should read "...by a long-lasting hyperpolarization.." [i.e., singular].
- p. 9, line 10 from below, "our previous data" should be replaced by "the data shown above"
- p. 9, line 7 from below: "intact" rather than "endogenous"
- p. 10, line 11: Egaba cannot obviously be given in mM. Please rewrite: "...a [Cl_i] value of 8.4 mM was calculated based on Egaba." (please see also point k) above)
- p. 13, line 8 from below: Fig 7c-d is current clamp data, please remove "currents" in "AHP currents following"
- p. 14: the abbreviation LTS is used only once after this and is therefore obsolete

Reviewer #2 (Remarks to the Author):

In the present paper Ha and co-worker present a new and unexpected mechanistic basis for the phenomenon of spike frequency adaptation in neurons. The authors combine electrophysiological recordings, specific pharmacology, gene knock-down strategies, expression analysis and behavioral tests to identify the Ca²⁺-activated Cl⁻ channel anoctamin-2 (ANO2) as the major basis for spike frequency adaptation in thalamocortical (TC) neurons of the ventrobasal thalamic complex (VB). This finding is novel and of high interest for the field of neuroscience. In general the paper is technically sound and well written. The data presented appear to be of high quality.

Nevertheless there are some shortcomings which dampen my enthusiasm.

1) It would be important to evaluate to what extend the ANO2-dependent mechanism shown here is a general mechanism of spike frequency adaptation in neurons. Is it specific for VB or thalamus? Is it present in all adapting neuronal cell types? Since TC neurons in different nuclei and species usually show only moderate spike frequency adaptation [1-3], it would be essential to see whether the results are representative for TC neurons from other thalamic nuclei and more importantly for neurons known for prominent spike frequency adaptation, like hippocampal pyramidal cells (which even stop firing during long depolarizing steps).

- 2) It has been shown that the postnatal action potential firing maturation has a slow time course with TC neurons from 3-7 month old mice revealing firing rates up to 200 Hz for strong (700 pA) depolarizations [3]. Therefore complete input / output curves should be shown and older animals should be tested.
- 3) The regulation of tonic firing by ANO2 in response to continuous current injection has been analyzed here. This form of depolarization may recruit Ca²⁺ sources in a different way compared to synaptic stimulation. It should be therefore determined whether ANO2 is also influencing the response to repeated phasic excitatory postsynaptic potential (EPSP)-like stimuli [3].
- 4) In the mammalian brain, external Ca²⁺ concentrations range from 1.5 to 2.0 mM [4]. In the present paper an elevated extracellular Ca²⁺ concentration was used. It should be tested whether a more physiological concentration (like 1.8 mM) sustains this new mechanism. This is interesting since the authors point out several times that TC neurons have to be highly activated to see the effect. This may therefore be a mechanism which is also relevant for pathophysiological conditions in the brain.
- 5) The mechanisms of tonic firing and spike frequency adaptation in TC neurons are complex and have been addressed to some degree before. It has been shown that Kv3.2, Kv1, SK, BK, KCNQ and T-type Ca²⁺ channels play an important role thereby pointing to the contribution of additional mechanisms [3,5-7]. While this not fully reflected in the discussion of the paper, the results presented here may also point in this direction. Knockdown of ANO2 increased the adaptation index from 50% to 77% (difference of 27%) thereby leaving another 23% open for additional (partially Ca²⁺-independent) mechanisms. Therefore other mechanisms like BK channels may be tested. Although iberiotoxin is not influencing the Ca²⁺-dependent tail current, spike frequency adaptation may be decreased. In addition the experiments shown in Fig. 3e may indicate that a single experiment in which apamin decreased the adaptation index may have prevented significant results. The number of experiments should be increased to address this possibility.
- 6) For the behavioral tests the injection sites should be documented for all recordings. Were the effects different for VPL (ventral posterior lateral nucleus) and VPM (ventral posterior medial nucleus) hits since both regions of the VB serve slightly different functions? It should be noted that also a reduction of specific high-frequency bursting in TC neurons of the VB may be associated with increased pain responses [7-9]. In this respect it would be interesting to see whether the intra-burst frequency of spikes (cf., Fig. 7j) was altered by ANO2 knock-down.
- 7) The membrane properties of TC neurons should be described in more detail (e.g., resting membrane potential, input resistance, membrane capacitance). What were the criteria for including cells into the statistical analysis? Was there any delayed onset of firing, since this is a characteristic feature of TC neurons in different species? In addition more methodological details should be stated. How was liquid junction potential correction done? Was this correction different for different solutions? What is the expected intracellular Ca²⁺ concentration for the used recording solutions? What was the recording temperature? Which test was used to determine normal distribution of data points? Which tests were performed to assess antibody specificity?

Additional points

P. 11: What parts of the "TC region" is referred to here?

P. 17: The lack of recurrent oscillatory bursting in TC neurons in the present study may be explained by the orientation (which is not stated in the method section) of the slices used for recordings. Typically only horizontal thalamic slices reveal recurrent oscillatory bursting [10].

P. 18: How is the Ca²⁺-sensitivity of ANO2 compared to SK and BK channels?

P. 20: Is there a contribution of ANO2 to the resting membrane potential?

References

- 1 McCormick, D. A. & Pape, H.-C. Acetylcholine inhibits identified interneurons in the cat lateral geniculate nucleus. *Nature* 334, 246-248 (1988).
- 2 Williams, S. R., Turner, J. P., Anderson, C. M. & Crunelli, V. Electrophysiological and morphological properties of interneurons in the rat dorsal lateral geniculate nucleus in vitro. *J. Physiol. (Lond.)* 490.1, 29-147 (1996).
- 3 Kasten, M. R., Rudy, B. & Anderson, M. P. Differential regulation of action potential firing in adult

murine thalamocortical neurons by Kv3.2, Kv1, and SK potassium and N-type calcium channels. *J Physiol* 584, 565-582 (2007).

4 Egelman, D. M. & Montague, P. R. Calcium dynamics in the extracellular space of mammalian neural tissue. *Biophys J* 76, 1856-1867 (1999).

5 Deleuze, C. et al. T-type calcium channels consolidate tonic action potential output of thalamic neurons to neocortex. *J Neurosci* 32, 12228-12236 (2012).

6 Ehling, P. et al. Ca(2+)-dependent large conductance K(+) currents in thalamocortical relay neurons of different rat strains. *Pflugers Arch* 465, 469-480 (2013).

7 Cerina, M. et al. Thalamic Kv 7 channels: pharmacological properties and activity control during noxious signal processing. *Br J Pharmacol* 172, 3126-3140, doi:10.1111/bph.13113 (2015).

8 Huh, Y. & Cho, J. Urethane anesthesia depresses activities of thalamocortical neurons and alters its response to nociception in terms of dual firing modes. *Frontiers in behavioral neuroscience* 7, 141, doi:10.3389/fnbeh.2013.00141 (2013).

9 Huh, Y. & Cho, J. Discrete pattern of burst stimulation in the ventrobasal thalamus for anti-nociception. *PloS one* 8, e67655, doi:10.1371/journal.pone.0067655 (2013).

10 Huntsman, M. M., Porcello, D. M., Homanics, G. E., DeLorey, T. M. & Huguenard, J. R. Reciprocal inhibitory connections and network synchrony in the mammalian thalamus. *Science* 283, 541-543 (1999).

Reviewer #3 (Remarks to the Author):

The manuscript by Ha et al describes some generally well conducted experiments that conclude that the modulation of action potential firing frequency is due to the activation of the calcium activated chloride channel ANO2. This finding is of significant interest as cation channels (e.g. calcium-activated potassium and HCN channels) typically modulate firing frequency, and this is the first demonstration that a chloride channels plays such a role in thalamic neurons. The manuscript is well written and it usually easy to follow the logic of each experiment. In general the data appear to be of good quality and the conclusions sound.

I have some, mostly minor, comments and suggestions.

1. Details should be added to indicate the extent of the region infected with AAV. This will be important for interpretation of the behavioral experiments, as spread of the virus could weaken the conclusion that the behavioral differences resulted from knockdown of ANO2 in thalamic neurons. The inset in Figure 8a showing mCherry staining is not described or discussed in the results text. The percentage of thalamic neurons expressing mCherry should be given with an indication of the numbers/percentage of ANO2-expressing cells infected by the virus.
2. It would be instructive to determine the behavioral effect of ANO2 knockdown on an acute pain stimulus (e.g. hot plate or tail flick) that has a short duration where repetitive neuronal firing may be less evident.
3. Standard controls should be used for the immunohistochemistry - staining should be shown for no ANO2 primary antibody (in e.g. Figure 6c). How specific is the antibody staining for neurons? Are there data with this antibody from ANO2-null mice?
4. Some of the IHC staining is difficult to see (e.g. Best1). Can the images be improved?
5. Was the effect of removing calcium from the external solution on spike frequency reversible (page 5/6, Fig 1a) or is this an irreversible effect?
6. As niflumic acid can act on some cation channels the conclusion that the NFA inhibited current is an anion current (page 7) is premature. This is only confirmed when the reversal potentials are measured (page 8)
7. The statement (page 17) that ANO2 is localized to the soma of thalamic neurons, unlike SK channels, needs clarification. Does this mean that ANO2 is not expressed on dendrites? ANO2 has been reported to be targeted to dendrites in Purkinje neurons (Zhang et al 2015 PLoS One 10: e0142160).
8. The hyperpolarizing protocol used for Figure 7k should be described in the methods (hyperpolarized to what membrane potential for what time period). At present there is a comment

in the figure legend about hyperpolarizing to $\sim -80\text{mV}$ but the results section mentions 'the hyperpolarizing protocol).

9. What is the free calcium concentration in the internal solution with 5Ca-EGTA-NMDG?
10. State the supplier for the Mini Analysis software.
11. A GFAP antibody was used but details are not given in the methods section.
12. Define mANO1 and mANO2 as mouse ANO1 and ANO2
13. Define APA when first introduced (page 8)
14. Page 18. "...actively emits Cl^{sup} ."
15. Extrudes or effluxes would be a more appropriate word than 'emit'

Reponses to Reviews

Reviewers' comments:

Reviewer #1 (Remarks to the Author):

This is an interesting and potentially important study in its field. The authors show that spike-frequency adaptation in thalamocortical neurons is mediated by the Ca^{2+} -activated Cl^- channel (CACC) anoctamin-2 (ANO2). This finding is (to the best knowledge of the present referee) a novel one. However, the paper has shortcomings which should be carefully addressed by the authors. Most of these can be overcome by text revisions, but perhaps one experiment could be added as explained in the first point in the list below.

A: We thank the reviewer for positive and insightful comments on our manuscript and for constructive suggestions to strengthen the manuscript. We tried our best to answer all inquiries and concerns that were raised by the reviewer and edited the figures and manuscript to address the suggestions. The edited text in the revised manuscript is outlined in blue for clarity. Our specific responses to each point that was raised by the reviewers are also outlined below in blue for clarity.

Major comments

1) Given the key role of intracellular Cl^- (Cl_i), it would be interesting if the authors would provide data on spike-frequency adaptation (SFA) under conditions where Cl_i would be intact and prone to activity-dependent variation. The gramicidin patch technique has been used only in experiments using GABA to probe the steady-state level of Cl_i and the resulting E_{Cl} . For studying SFA under conditions with intact Cl_i , on-cell patching might also be an effective approach. Anyway, it is entirely possible that even under in vitro conditions, somatic Cl_i would show activity-dependent (spike-firing dependent) fluctuations and thereby modulate both the ANO2-mediated current and, consequently, SFA. I do not insist on this experiment, but this is a point that should at least be taken up in the Discussion.

A. We thank the reviewer for the suggestions to strengthen the main idea of our manuscript by confirming the ANO2-mediated spike frequency adaptation under conditions with intact $[\text{Cl}^-]_{\text{in}}$. In response to the reviewer's suggestion, we performed the gramicidin-perforated patch clamp to measure the ANO2-mediated spike frequency adaptation, although the reviewer did not insist on the experiment. This was done on TC neurons that were infected with either AAV-shANO2 or AAV-Scr. Then we compared the spike frequency adaptation of those TC neurons in which $[\text{Cl}^-]_{\text{in}}$ could be intact. Perforated patch recordings showed that the knockdown of ANO2 increased spike frequency adaptation index in the condition of intact $[\text{Cl}^-]_{\text{in}}$ (see below), indicating that SFA disappeared with knockdown of ANO2. This result was added to the revised manuscript (Fig. 7k-l, Page 14 line 21 in the revised text).

2) Surprisingly, the authors do not clearly state the major, general novelty of their work in the Abstract. On p. 9 they state that "no previous study has described a CACC that is involved in mAHP currents and neuronal spike patterns in the central nervous system". This important point should be included into the Abstract.

A. We thank the reviewer for this positive comment on the significance of our manuscript. In response to the reviewer's comment, we modified the abstract to state the novelty of our work as suggested by the reviewer. (Page 2 line 11 in the revised text)

3) The adaptation index used presently will be extremely sensitive to minor changes in the first and last interspike interval. I would suggest using an average of at least two if not three ISIs (i.e., the initial 2 or 3 and the last 2 or 3 ISIs) two make the parameter more robust.

A. We agree with the reviewer on the point that calculation of the adaptation index could be sensitive to the small changes in the first and last ISIs. In response to the reviewer's suggestion, we adjusted the analysis method of the adaptation index to use an average of the first two and the last two ISIs and newly analyzed results were added to the revised manuscript. We also revised the text as "The adaptation index (AI) was obtained by dividing the average of the first 2 ISIs by average of the last 2 ISIs." and replaced. (Revised Fig. 1d, 3d, 7f. Page 6 line 3 in the revised text)

4) The authors should explain why pain responses only have been studied (section starting on p. 14). In thalamocortical sensory functions, other sensory modalities take a much more central place in the literature.

A. We apologize for the insufficient explanation on why we conducted visceral pain test. The reason is that it was previously reported that the altered firing properties of VB thalamic functions could influence the visceral pain responses¹. Based on this, we predicted that the pain response could be a good tool to test the thalamic function affected by the changes in spiking frequency of TC neurons (increased AHP and decreased spike frequency adaptation). Here we measured pain responses to prove our idea that changes in the firing frequency adaptation in TC neurons can modulate the thalamic function as ANO2 is highly expressed in VB nuclei in TC region where the pain signals are

known to be relayed. In general, the sensory information from pain and temperature receptors on peripheral body is transmitted through the spinothalamic tract. The spinothalamic tract is a representative sensory pathway from the periphery to the thalamus, specifically ventroposterior medial (VPM) and lateral (VPL) nucleus. Although VB thalamic function is closely related with pain behavior, we agree with the reviewer's comments that other sensory modalities could be affected, too. In the future, it would be very interesting to study the function of ANO2 in modulating other sensory modalities. In response to the reviewer's suggestion, this rationale was added in the discussion part (Page 20 line 10 in the revised text).

5) Discussion, start of: the data in Fig. 2d do not indicate that this is an outwardly rectifying current as stated in the text.

A. We apologize to the reviewer for the confusion from mismatched text. The data showing the outwardly rectifying current characteristic of ANO2 is Fig. 5e, not Fig. 2d. We corrected the discussion in the revision and specified the outwardly rectifying characteristic of current in the discussion data (Page 18 line 22 in the revised text).

6) Ca²⁺ buffering in the pipette solution has been done with very low and variable amounts of EGTA. Because somatic Ca²⁺ is a key player, the authors should provide information on the possible effects (or attempts to minimize them) of the exogenous buffers. This would be relevant for the discussion on p. 19, 2nd paragraph.

A. As the reviewer mentioned, it was important not to disturb the intracellular Ca²⁺ while we measured Ca²⁺-activated current and Ca²⁺-mediated firing frequency adaptation. Based on this, we used very low levels of EGTA to minimally chelate the Ca²⁺ contamination from other chemicals and maintain the same level of free Ca²⁺ level in the intrapipette solution. We used 0.02 mM EGTA in the experiments to measure spike firing rates and used 0.1 mM EGTA to measure the Ca²⁺ activated AHP current. This low EGTA concentration in intrapipette solution was often used by many studies to study the Ca²⁺-current-mediated changes in AHP current or firing pattern changes with minimal disturbance onto the intracellular Ca²⁺ level.^{2,3,4}

7) p. 18: Outward rectification has nothing to do with a channel's capability to hyperpolarize the plasma membrane. While this kind of a misconception is frequent, it is the reversal potential only which dictates the polarity of a channel-mediated response.

A. Although we agree with the reviewer that outward rectification has nothing to do with a channel's capability to hyperpolarize the plasma membrane, we never stated that outward rectification of ANO2 contributes to hyperpolarization. Instead, we stated that outward rectification limits the ability for neurons to depolarize ("Therefore, ANO2 would generate relatively large outward currents in TC

neurons with an endogenous ionic content at the depolarized membrane potential.”, Page 18 line 15, Discussion). We believe that outward rectification can act as a ceiling for neurons not to go beyond a certain depolarized membrane potential due to ANO2. We also agree with the reviewer on the comment that the reversal potential with endogenous intracellular chloride ion concentration would be the most important factor to determine the hyperpolarizing function of ANO2. Therefore, we first discussed the reversal potential with endogenous intracellular chloride ion concentration first and then addressed the outward rectification in the revised Discussion part. (Page 18 line 22 in the revised text)

8) There's a major confusion with regard to Fig. 7 where, for instance, panels j and k are not even mentioned in the Fig. legend. Please also cross-check with main text (p. 13).

A. We are sorry for the confusion of the missing description of Fig. 7 about the burst firing of TC neurons. We included a description and legend for the figure in the revised manuscript. (Fig. 7i-j, figure legend in Page 43 line 8)

9) The style of writing is verbose and sometimes the flow is somewhat clumsy. I would urge the authors to aim at a much more compact style of writing throughout the ms (especially the Results section). This would also make the paper shorter.

A. We took the reviewer's comments into account and modified the word choice, notably the result section. We revised this section explaining the rationale to do each experiment in a more concise manner and brought our focus more to the meaning of the results. We also substantially revised the description of the behavioral test. (Page 15 line 12 in the revised text)

Further comments

a) The title should state which cells are being studied

A. In response to the reviewer's suggestion, we modified the title to “The Ca²⁺-activated chloride channel anoctamin-2 mediates spike-frequency adaptation and regulates sensory transmission in thalamocortical neurons”. (Page 1 line 1 in the revised text)

b) The running title is not well-formulated. I would suggest "ANO2 and spike frequency adaptation" or something along these lines ("dampening information transfer" is vague/ambiguous)

A: As per the reviewer's suggestion, we modified the running title to “ANO2-mediated spike-frequency adaptation regulates information transfer from thalamus to cortex”. (Page 1 line 5 in the revised text)

c) p. 4: "is still debated". Here the authors cite papers from 2001-2003. Novel references are needed to justify "still debated".

A. We agree with the reviewer's suggestion. We additionally supplemented a reference article published recently (2014). (Page 4 line 2)

d) p. 6, line from below: "holding current" should be "holding potential"

A. We apologize to the reviewer for the incorrect expression and corrected the sentence as suggested. (Page 6 line 22)

e) p. 6, last line: these were surely not "0 mV prepulse steps" but rather "prepulse steps to 0 mV"

A. We corrected the sentence as suggested by the reviewer in the revised manuscript. (Page 7 line 4)

f) p. 7 line 2: What are the error bars in the text and Figures? SD or SEM? I am not sure whether this information has been provided. Anyway, the numerical value 32 ± 7.1 ms does not match with what is shown in Fig. 1j. Perhaps similar errors are present elsewhere - please check.

A. We apologize to the reviewer for the insufficient description. All the error bars in text and figures indicated standard error of the mean (SEM) and we indicated it in the revised manuscript (Page 7 line 6 and Page 25 line 23 in method section). In fact, Fig. 1j showed that the values of current decay time constants (τ) normalized with mean decay tau at 50 ms prepulse durations which was 32 ± 7.1 ms (averaged from the decay time constant of AHP current with 5 different prepulse duration). Therefore, the value of first bar (50 ms prepulse duration) in figure 1j is 1. These points was reflected to the revised manuscript included text and legend.

g) It is confusing to separate the upper panels of Fig. 1 into a and b, they are from the same cell.

A. We accepted the reviewer's comment and revised the figure numbers from Fig. 1a and 1b into Fig 1a with small roman numbering i) and ii). (Revised Fig. 1a)

h) The voltage protocol used should be illustrated in Fig. 2c (which also lacks calibration ms/pA), and in Fig. 7j.

A. In response to the reviewer's suggestion, we added the voltage protocol in Fig. 2c and Fig. 7i.

i) p. 9: "These results indicate that SK channels contribute substantially to the generation of Ca^{2+} -activated mAHPs, but are not critical to spike frequency adaptation." This sentence should be clarified (see capital letters below), perhaps like this: "These results indicate that SK channels contribute substantially to the generation of Ca^{2+} -activated mAHPs AND TONIC SPIKE FIRING RATE, but are not.."

A. We accepted the viewpoint of the reviewer to clarify the meaning of the result and modified the sentence as suggested by the reviewer. (Page 9 line 10)

j) p. 10, line 6: "...the intercept on..". Please state intercept with what - i.e. intercept with the voltage

axis.

A. We accepted the viewpoint of the reviewer to clarify the meaning of the result and modified the sentence as suggested by the reviewer. (Page 10 line 7)

k) The estimated 8.4 mM Cl_i has not been corrected for bicarbonate permeability. With an internal HCO_3^- of 15 mM typical for CNS neurons, and a relative permeability of 0.2 or 0.3 vs Cl^- , the real Cl_i based on the E_{gaba} measurements is around 5-6 mM.

The authors might also note that ANO2 is a very unspecific anion channel which may carry substantial (depolarizing) HCO_3^- current components (see Farrant and Kaila 2007 review for a quantitative treatment).

A. We thank the reviewer for the critical comments to allow us to strengthen the main findings of our manuscript. We corrected the endogenous $[Cl^-]_{in}$ to 5.4 mM with a correction for the bicarbonate permeability. In following, we corrected the endogenous $[Cl^-]_{in}$ in the results addressing the bicarbonate permeability issue with references (Page 10 line 13).

As the reviewer states, it was known that Ca^{2+} -activated Cl^- channels have the HCO_3^- permeability. The relative permeability of HCO_3^- versus Cl^- ($P_{HCO_3^-}/P_{Cl^-}$) of Human Bestrophin 1 (hBEST1) channel, one of the CACCs, is around 0.44⁵. Anoctamin 1 (ANO1) channel was reported to have 0.35 of $P_{HCO_3^-}/P_{Cl^-}$ ⁶. We included that CACCs are also permeable to HCO_3^- in the revised text. (Page 10 line 14)

l) p. 12, line 5: "remaining" should be "main".

A. We revised the sentence as suggested by the reviewer. (Page 12 line 10)

Minor comments

- Abstract line 7: "by an increased tonic spikes" - please correct.

A. We corrected the grammar in the sentence as suggested by the reviewer. (Page 2 line 7)

- p. 4, line 10 from below: "mAHP is are known" - please correct

A. We corrected the grammar in the sentence as suggested by the reviewer. (Page 4 line 13)

- p. 4, line 5 from below: to make it easier for the reader, please replace " Ca^{2+} activating mAHP currents" by " Ca^{2+} currents which activate mAHP currents"

A. We revised the sentence as suggested by the reviewer. (Page 4 line 17)

- p. 7, line 3: "elicited larger Ca^{2+} influxes" is not based on data; please reformulate.

A. We revised the sentence to "Prepulse steps from -60 mV to 0 mV could activate voltage-dependent Ca^{2+} channels to induce Ca^{2+} influx into the cell. Therefore, the durations of voltage steps regulating the amount of intracellular Ca^{2+} concentration were positively correlated with the amplitudes and lengths of the tail currents" as it has been known from previous studies. (Page 6 line 23)

- p. 8, line 3 from below should read "...by a long-lasting hyperpolarization.." [i.e., singular].
A. We corrected the grammar in the sentence as suggested by the reviewer. (Page 9 line 1)
- p. 9, line 10 from below, "our previous data" should be replaced by "the data shown above"
A. We revised the sentence as suggested by the reviewer. (Page 9 line 18)
- p. 9, line 7 from below: "intact" rather than "endogenous"
A. We edited the sentence as suggested by the reviewer. (Page 9 line 21)
- p. 10, line 11: E_{gaba} cannot obviously be given in mM. Please rewrite: "...a $[\text{Cl}^-]_i$ value of 8.4 mM was calculated based on E_{gaba} ." (please see also point k) above)
A. We corrected the expression on intracellular Cl^- level and E_{GABA} as commented by the reviewer. We also corrected the endogenous $[\text{Cl}^-]_i$ to 5.4 mM with the bicarbonate permeability according to the comment (k). (Page 10 line 13)
- p. 13, line 8 from below: Fig 7c-d is current clamp data, please remove "currents" in "AHP currents following"
A. We modified the sentence as suggested by the reviewer. (Page 13 line 21)
- p. 14: the abbreviation LTS is used only once after this and is therefore obsolete
A. We removed the abbreviation from the text as suggested by the reviewer (Page 14 line 16).

Reference

1. Kim D, *et al.* Thalamic control of visceral nociception mediated by T-type Ca²⁺ channels. *Science* **302**, 117-119 (2003).
2. Cueni L, *et al.* T-type Ca²⁺ channels, SK2 channels and SERCAs gate sleep-related oscillations in thalamic dendrites. *Nature neuroscience* **11**, 683-692 (2008).
3. Jones SL, Stuart GJ. Different calcium sources control somatic versus dendritic SK channel activation during action potentials. *The Journal of neuroscience : the official journal of the Society for Neuroscience* **33**, 19396-19405 (2013).
4. Bond CT, *et al.* Small conductance Ca²⁺-activated K⁺ channel knock-out mice reveal the identity of calcium-dependent afterhyperpolarization currents. *The Journal of neuroscience : the official journal of the Society for Neuroscience* **24**, 5301-5306 (2004).
5. Qu Z, Hartzell HC. Bestrophin Cl⁻ channels are highly permeable to HCO₃⁻. *American journal of physiology Cell physiology* **294**, C1371-1377 (2008).
6. Jung J, Nam JH, Park HW, Oh U, Yoon JH, Lee MG. Dynamic modulation of ANO1/TMEM16A HCO₃⁻ permeability by Ca²⁺/calmodulin. *Proceedings of the National Academy of Sciences of the United States of America* **110**, 360-365 (2013).

Responses to Reviewer2

Reviewers' comments:

Reviewer #2 (Remarks to the Author):

In the present paper Ha and co-worker present a new and unexpected mechanistic basis for the phenomenon of spike frequency adaptation in neurons. The authors combine electrophysiological recordings, specific pharmacology, gene knock-down strategies, expression analysis and behavioral tests to identify the Ca²⁺-activated Cl⁻ channel anoctamin-2 (ANO2) as the major basis for spike frequency adaptation in thalamocortical (TC) neurons of the ventrobasal thalamic complex (VB). This finding is novel and of high interest for the field of neuroscience. In general the paper is technically sound and well written. The data presented appear to be of high quality. Nevertheless there are some shortcomings which dampen my enthusiasm.

A: We thank the reviewer for the kind and insightful comments on our manuscript. We are especially grateful to the reviewer for many constructive suggestions to allow us to strengthen the manuscript. We have tried our best to answer all inquiries and concerns raised by the reviewer and edited the figures and manuscript to address these suggestions. Here, we have done all the experiments suggested by the reviewer. The edited text in the revised manuscript is outlined in blue for clarity. Our specific responses to each point raised by the reviewers are also outlined below in blue for clarity.

1) It would be important to evaluate to what extent the ANO2-dependent mechanism shown here is a general mechanism of spike frequency adaptation in neurons. Is it specific for VB or thalamus? Is it present in all adapting neuronal cell types? Since TC neurons in different nuclei and species usually show only moderate spike frequency adaptation^{1, 2, 3} it would be essential to see whether the results are representative for TC neurons from other thalamic nuclei and more importantly for neurons known for prominent spike frequency adaptation, like hippocampal pyramidal cells (which even stop firing during long depolarizing steps).

A. We appreciate the reviewer for the comments concerning the scope of our manuscript. In this study, we examined how the spike frequency adaptation is generated in thalamic neurons in the VB nuclei (included VPM and VPL), that relay somatosensory information. Unfortunately, we have not specified sub-region of thalamus in the current study (it might be interesting to the future work). Nevertheless, as the reviewer suggested, we tested the spike frequency adaptation in hippocampal neurons to see if ANO2-mediated spike frequency adaptation could be a general mechanism. As shown in the figure below, CA1 pyramidal neurons infected with AAV-*shANO2* displayed the reduced spike frequency adaptation compared to CA1 neurons infected with AAV-*Scr*, though a larger portion of spike frequency adaptation remained. These results suggest that other channels such as potassium channels as known in many previous studies contribute a lot to the spike adaptation in CA1 neurons⁴.

5. We hope that this preliminary data would answer some of the questions raised by the reviewer. We also hope that the reviewer would generously understand that we will not include these results in this manuscript because this is not the scope of our current work.

Because we studied the spike frequency adaptation of the thalamocortical neurons, not other regions of brain, we limited the mechanism of spike frequency adaptation to thalamic neurons. This result was reflected in the revised version of manuscript with a new title, “Ca²⁺-activated chloride channel anoctamin-2 mediates spike frequency adaptation and regulates sensory transmission in thalamocortical neurons”. (Page 1 line 1)

- Hippocampal CA1 neurons recording -

2) It has been shown that the postnatal action potential firing maturation has a slow time course with TC neurons from 3-7 month old mice revealing firing rates up to 200 Hz for strong (700 pA) depolarizations³. Therefore complete input / output curves should be shown and older animals should be tested.

A. We absolutely agree with the reviewer on the comment that the input-output curve of firing rate in TC neurons should be shown to strengthen our data. In response to the reviewer’s comment, the complete input-output curves measured from TC neurons in 5-6 weeks old mice were added to the revised Fig. 1e and also in the revised text (Page 6 line 7).

We also accepted the reviewer's suggestion to measure the firing rate in older animals and measured spike frequency adaptation of TC neurons from 4-month-old mice. Input-output curves of spike firing rates, comparing between 5~6-week-old and 4-month-old mice, have been added to the revised supplementary Fig. S2. TC neurons from older mice also showed firing frequency adaptation pattern similar to 5-6 week old TC neurons (Suppl. Fig. S2a-b) with similar adaptation index (Suppl. Fig. S2c). Input-output curve showed that there is no significant difference in spike firing rates of TC neurons from 5-week-old and 4-month-old mice with depolarizing current steps up to 400 pA. The stronger depolarizing current steps ranging from 500 to 700 pA induced a further increase in firing frequency in TC neurons from 4-month-old mice whereas TC neurons from 5-week old mice could not increase the firing frequency substantially. This tendency agrees with the previous report³. We included this reference in the revised manuscript (revised Suppl. Fig. S2).

3) The regulation of tonic firing by ANO2 in response to continuous current injection has been analyzed here. This form of depolarization may recruit Ca^{2+} sources in a different way compared to synaptic stimulation. It should be therefore determined whether ANO2 is also influencing the response to repeated phasic excitatory postsynaptic potential (EPSP)-like stimuli³.

A. We appreciate the reviewer for suggesting an experiment to apply repeated phasic EPSP-like stimuli. To apply EPSP-like stimuli, we generated a model EPSC using double exponential equation as shown in equation

$$(1) A(t) = A_1 e^{\left(\frac{-t}{\tau_{rise}}\right)} - A_2 e^{\left(\frac{-t}{\tau_{decay}}\right)} \text{ and } (2) t_{peak} = t_0 + \frac{\tau_{decay}\tau_{rise}}{\tau_{decay}-\tau_{rise}} \ln\left(\frac{\tau_{decay}}{\tau_{rise}}\right).$$

Rising and decay tau were 0.5 and 4.07, respectively⁶, adopted from as shown below (Suppl. Fig. S3a). EPSP-like stimuli were delivered to TC neurons by applying various frequencies (20, 50 and

100 Hz) and amplitudes (ranging from 200 to 1000 pA) of the modeled EPSC. We quantified the adaptation index injecting EPSP-like stimulation of 100 Hz in supplementary figure S3. A more detailed method is written for EPSP-like stimulation in the method section. TC neurons infected with AAV-*shANO2* displayed a tendency of an increase of adaptation index value compared to those infected with AAV-*Scr* with an insignificant p-value. (Suppl. Fig. S3c, $p=0.07$). Also in 20 Hz and 50 Hz, the spike adaptation was prominent as shown below and the firing pattern was similar to previous work³, implying that firing frequency adaptation may occur responding to physiological excitatory inputs.

100 Hz stimulation (Suppl. Fig. 3)

20 and 50 Hz stimulation

20Hz EPSC-like stim.

50Hz EPSC-like stim.

4) In the mammalian brain, external Ca^{2+} concentrations range from 1.5 to 2.0 mM ⁷. In the present paper an elevated extracellular Ca^{2+} concentration was used. It should be tested whether a more physiological concentration (like 1.8 mM) sustains this new mechanism. This is interesting since the authors point out several times that TC neurons have to be highly activated to see the effect. This may therefore be a mechanism which is also relevant for pathophysiological conditions in the brain.

A. In response to the reviewer's comment, we executed the experiments to measure spike frequency adaptation under more physiological Ca^{2+} concentration (1.5 - 2.0 mM in the brain). Under the extracellular buffer containing 1.8 mM Ca^{2+} concentration, we obtained a similar value of adaptation index from TC neurons shown in revised Suppl. Fig. S1. This result suggested that the phenomenon of spike adaption is also shown under physiological conditions (Revised Suppl. Fig. S1 and Page 5 line 16 in the revised text).

“TC neurons have to be highly activated to see the effect” means that the Ca^{2+} mediated spike frequency adaptation is more distinctively visible when TC neurons make multiple spikes. Therefore, we believe that this spike frequency adaptation works as a type of suppression of spike generation at high frequency when there are many of excitatory inputs, which could be persistent pain stimuli, to TC neurons in the brain.

5) The mechanisms of tonic firing and spike frequency adaptation in TC neurons are complex and have been addressed to some degree before. It has been shown that Kv3.2, Kv1, SK, BK, KCNQ and T-type Ca^{2+} channels play an important role thereby pointing to the contribution of additional mechanisms^{3, 8, 9, 10}. While this not fully reflected in the discussion of the paper, the results presented here may also point in this direction. Knockdown of ANO2 increased the adaptation index from 50% to 77% (difference of 27%) thereby leaving another 23% open for additional (partially Ca^{2+} -independent) mechanisms. Therefore other mechanisms like BK channels may be tested. Although iberiotoxin is not influencing the Ca^{2+} -dependent tail current, spike frequency adaptation may be decreased.

A. We absolutely agree with the reviewer and were aware of these previous studies. We examined the possibility that many kinds of potassium channels and T-type Ca^{2+} channels could affect this

phenomenon. Firstly, as we showed in Fig 7i-j, the burst firing of TC neurons did not change in the condition of knockdown of ANO2 by AAV-*shANO2*, indicating T-type Ca^{2+} channels were not altered in the condition and minimally contribute to the spike adaptation at this condition. Next, we tested Iberiotoxin on the Ca^{2+} -activated tail current as the reviewer mentioned. In fact, it did not affect either the Ca^{2+} -activated tail current or tonic firing frequency adaptation, indicating BK channels may not contribute to spike frequency adaptation. This result is now reflected in to the revised text (Page 8 line 1). Although we did not test all the other potassium channels, while the tonic firing frequency adaptation seemed to be greatly affected by the Ca^{2+} -activated Cl^- channel component. However, despite of our some control data, we agree with the reviewer's consideration that we cannot completely exclude the possibility of the other channels' contribution to firing frequency adaptation. To respond the reviewer's comment, we discussed this possibility in the revised manuscript of discussion section (Page 17 line 23)

In addition the experiments shown in Fig. 3e may indicate that a single experiment in which apamin decreased the adaptation index may have prevented significant results. The number of experiments should be increased to address this possibility.

A. We fully agree with the reviewer for the concerning on the influence of dissimilar data point. To re-evaluate our statistics, first, we applied a new calculation method for analysis of the adaptation index in that we used "average of the first 2 ISIs by average of the last 2 ISIs" (Page 6 line 3) to exclude the possibility that minor changes in either the first ISI or the last ISI as suggested by another reviewer. Next, we reexamined the effect of apamin because some of the paired recordings showed an increase in adaptation index whereas some did not. We recalculated the adaptation index with or without the specific single experiment which showed the decrease in adaptation index (0.46 ± 0.04 vs. 0.52 ± 0.06 , $p=0.22$ by paired t-test, $n=6$ from 3 mice). Indeed, there was no change of significance with or without the data point indicating the contribution of SK channel to tonic spike adaptation is not significant. This new method for analysis of adaptation index and re-evaluation of statistics was reflected in the revised manuscript.

6) For the behavioral tests the injection sites should be documented for all recordings. Were the effects different for VPL (ventral posterior lateral nucleus) and VPM (ventral posterior medial nucleus) hits since both regions of the VB serve slightly different functions?

A. In response to the reviewer's comment, we provided schematic diagram of injection sites of the AAV virus from the brains of all mice used for the visceral pain test (Suppl. Fig. S7a). Furthermore, we displayed an example for the extent of the AAV-infected region for each animal measured by the post-hoc staining of the AAV-infected area (mCherry-expressing area) to total VB region (Supple Fig. S7b). We confirmed that the broad VB region, including both the VPL and VPM, was covered by $70.2 \pm 3.9\%$. We excluded the mice whose AAV-infected VB region is below 50% because, as you could see in the viral infection area (S7b), single injection of virus infected both the VPL and VPM areas. We

could not distinguish the different effect of VPL vs. VPM. The reviewer's comment on sub-regional effect on behavior would be very interesting for future work.

It should be noted that also a reduction of specific high-frequency bursting in TC neurons of the VB may be associated with increased pain responses^{8, 11, 12}. In this respect it would be interesting to see whether the intra-burst frequency of spikes (cf., Fig. 7j) was altered by ANO2 knock-down.

A. In the response to the reviewer's concern whether or not intra-burst frequency of spikes was altered by ANO2 knock-down, we measured the intra-burst spike numbers and the intra-burst frequency under the condition of viral infection (AAV-Scr and AAV-shANO2). Because we were already aware of previous literatures^{8, 11, 12} indicated by the reviewer, we showed the burst firing in TC neurons in pain response. As a result, there was no difference between the two groups. These points were added in the revised manuscript (Fig. 7i-j).

7) The membrane properties of TC neurons should be described in more detail (e.g., resting membrane potential, input resistance, membrane capacitance). What were the criteria for including cells into the statistical analysis?

A. We agree with the reviewer on the comment and added Supplementary Table 1 showing the membrane properties of TC neurons. Neurons with series resistance < 30MΩ were accepted for statistical analysis (Page 25 line 22). Cell membrane resistance and capacitance were measured at 5mV depolarizing step. The resting membrane potential in whole cell configuration was measured in the current-clamp mode with zero current injection. This result was added to the revised manuscript and Supplementary table 1.

	2.4mM [Ca ²⁺]	1.8mM [Ca ²⁺]	AAV-Scr	AAV-shANO2	4-month-old
RMP(mV)	-67.1±1.3	-68.0±1.41	-66.7±1.15	-66.9±1.06	-64.4±1.3
C _m (pF)	147.3±13.9	155.2±15.6	120.58±6.71	123.27±5.39	165.2±13.4
R _{in} (MΩ)	214.1±37.0	264.3±60.8	238.66±32.93	257.33±19.7	235.3±31.6

RMP : resting membrane potential, C_m : membrane capacitance, R_{in} : input resistance

Was there any delayed onset of firing, since this is a characteristic feature of TC neurons in different species?

A. We did not observe any delayed onset of firing in mouse TC neurons.

In addition more methodological details should be stated. How was liquid junction potential correction done? Was this correction different for different solutions?

A. We thank the reviewer for the comment on junction potential. Through experimentation, we corrected the junction potential for each intrapipette (i.p.) solution and extracellular buffer pair. Therefore, the corrected junction potential values are different in each i.p. solution which can be found in the Fig. 2d data. (For example, mean value of junction potential in 60mM KCl intracellular pipette with aCSF (extracellular buffer) containing 2.4mM Ca²⁺ was 0.813 mV). This was described in the methods section of the revised manuscript.

What is the expected intracellular Ca²⁺ concentration for the used recording solutions?

A. We used an intracellular pipette solution with low amounts of EGTA (0.02 mM and 0.1 mM) to minimally chelate the Ca²⁺ contamination from other chemicals and measured Ca²⁺-activated current and Ca²⁺-mediated firing frequency adaptation. Although we could not directly measure the [Ca²⁺]_{in}, quantitative relationship between influx of Ca²⁺ by spike generation, voltage-gated Ca²⁺ channel and free Ca²⁺ concentration could be estimated as shown in the calcium-EGTA table and graphs calculated. We calculated these values based on the method for computing ion concentration¹³. The pH and temperature used to calculate were matched with the recording conditions (pH = 7.3, temperature = 25°C). We calculated the free [Ca²⁺]_{in} concentration with EGTA concentrations used for spike recording and current recording (see the table and graphs below) by adopting the range of total Ca²⁺ concentration based on the expected Ca²⁺ influx into neurons¹⁴. We calculated at free Ca²⁺ level from 0.01 μM to 10 μM of total Ca²⁺ because the intracellular Ca²⁺ concentration is 0.1 μM at rest, but increases up to 1 μM when cells are activated¹⁵.

	Total Ca (μM)	0.01	0.1	1	2	3	4
Free Ca (μM)	EGTA 0.02 mM	9.2E-05	9.2E-04	9.7E-03	2.0E-02	3.2E-02	4.6E-02
	EGTA 0.1 mM	1.9E-05	1.9E-04	1.9E-03	3.8E-03	5.7E-03	7.7E-03

	Total Ca (μM)	5	6	7	8	9	10
Free Ca (μM)	EGTA 0.02 mM	6.1E-02	7.8E-02	9.8E-02	1.2E-01	1.5E-01	1.8E-01
	EGTA 0.1 mM	9.8E-03	1.2E-02	1.4E-02	1.6E-02	1.8E-02	2.1E-02

What was the recording temperature?

A. Temperature during performing electrophysiological experiments was room temperature, which was about 25°C. We have added this information in the revised manuscript. (Page 22 line 13)

Which test was used to determine normal distribution of data points?

A. We used Ryan-Joiner normality test to check whether data points have normal distribution to be suitable for t-test. This test assesses normality by calculating the correlation between original data and the normal scores of data, and the data points are seemed to be normal when the correlation coefficient is near 1.

Which tests were performed to assess antibody specificity?

A. We confirmed specificity of ANO2 antibody by performing western blot and immunohistochemistry. In the western blot, we observed positive ANO2 signal in olfactory epithelium which is a known region of ANO2 expression, and ANO2-overexpressing NIH/3T3 cells, but not in normal NIH/3T3 cells (Figure 6a and 6b). In immunohistochemistry, we observed an ANO2 signal in AAV-Scr-infected TC neurons, but the signal was significantly reduced in AAV-shANO2-infected TC neurons (Figure 6d and 6e) indicating that the ANO2 antibody shows high specificity for the ANO2 protein. In fact, this antibody was provided by the research group which made ANO2 knockout mice and tested this antibody to the knockout mice¹⁶. Also, we examined the specificity of the secondary antibody to test its non-specific binding and obtained the supplementary figure S6 which indicates that no signal was observed in immunostaining data when the primary antibody was not used (Suppl. Fig. S6). We wish that the reviewer would agree with the antibody specificity from the results of several approaches and added results in the revised manuscript.

Additional points

P. 11: What parts of the "TC region" is referred to here?

A. We are sorry for the confusion. In fact, 'TC region' implied VB nuclei of the thalamus where we recorded electrophysiological data. To prevent the misreading, we replaced "TC region" to "VB nuclei in TC region" on page 11 line 7.

P. 17: The lack of recurrent oscillatory bursting in TC neurons in the present study may be explained by the orientation (which is not stated in the method section) of the slices used for recordings. Typically only horizontal thalamic slices reveal recurrent oscillatory bursting¹⁷.

A. We apologize for the lack of information in the method section.

1) We obtained horizontal thalamic slices for patch clamp recording and described this procedure in the methods section. (Page 22 line 5).

2) This is a misunderstanding in our data and the data in ¹⁷. The recurrent oscillatory activity (Fig. 3 in

¹⁷) is different from the bursting in TC neurons. This type of recurrent intra-thalamic oscillatory activity can be induced by electrically stimulating the internal capsule (IC) in horizontal slices as the reviewer commented. This recurrent activity can be induced only when there are intact reciprocal synaptic inputs between TRN and TC neurons in horizontal or thalamocortical (with 45-50 degree angular slice cutting) slices with simultaneous stimulation of a large population of thalamic neurons. Then, the synchronized inhibitory inputs from TRN neurons would silence TC neurons followed by a synchronized spiking activity which probably contains many burst spikes in TC neurons. But, we cannot differentiate the clustered spikes which look like bursting from real burst spikes because they were multiunit recordings using electrodes with low impedance.

3) Here we recorded the firing pattern from individual TC neuron with stimulation onto a single TC neuron in either whole-cell or gramicidine-perforated patch configurations. Therefore, we could not observe this recurrent activity, although we recorded in horizontal slices.

P. 18: How is the Ca^{2+} -sensitivity of ANO2 compared to SK and BK channels?

A.. ANO2 channel is known to need Ca^{2+} concentration higher than $1 \mu\text{M}$ for opening¹⁸ whereas ANO1 channels are activated at $0.1\text{-}0.3 \mu\text{M}$ Ca^{2+} concentration¹⁹. The Ca^{2+} sensitivity of ANO2 is known to be almost 10-fold lower than in ANO1 channel²⁰. BK and SK channels are activated at $10 \mu\text{M}$ and $0.3 \mu\text{M}$ Ca^{2+} concentration, respectively²¹. In summary, SK channels have the highest Ca^{2+} sensitivity. ANO2, and BK channels are next as an order (ANO1 = SK > ANO2 > BK).

P. 20: Is there a contribution of ANO2 to the resting membrane potential?

A. No, there is minimal contribution of ANO2 to the resting membrane potential. In fact, *AAV-shAno2* had no effect on the resting membrane potential as shown in the supplementary table 1 added to the revised version of the manuscript. We think that the activity of ANO2 in the resting membrane potential may be low because intracellular Ca^{2+} concentration is low at resting state and the reversal potential of Cl^- channels in TC neurons with endogenous $[\text{Cl}^-]_{\text{in}}$ was not far from the resting membrane potential as shown in Fig. 4.

References

1. McCormick DA, Pape HC. Acetylcholine inhibits identified interneurons in the cat lateral geniculate nucleus. *Nature* **334**, 246-248 (1988).
2. Williams SR, Turner JP, Anderson CM, Crunelli V. Electrophysiological and morphological properties of interneurons in the rat dorsal lateral geniculate nucleus in vitro. *The Journal of physiology* **490 (Pt 1)**, 129-147 (1996).
3. Kasten MR, Rudy B, Anderson MP. Differential regulation of action potential firing in adult murine thalamocortical neurons by Kv3.2, Kv1, and SK potassium and N-type calcium channels. *The Journal of physiology* **584**, 565-582 (2007).
4. Chen S, Benninger F, Yaari Y. Role of small conductance Ca(2+)-activated K(+) channels in controlling CA1 pyramidal cell excitability. *The Journal of neuroscience : the official journal of the Society for Neuroscience* **34**, 8219-8230 (2014).
5. Hirschberg B, Maylie J, Adelman JP, Marrion NV. Gating properties of single SK channels in hippocampal CA1 pyramidal neurons. *Biophysical journal* **77**, 1905-1913 (1999).
6. Miyata M, Imoto K. Different composition of glutamate receptors in corticothalamic and lemniscal synaptic responses and their roles in the firing responses of ventrobasal thalamic neurons in juvenile mice. *The Journal of physiology* **575**, 161-174 (2006).
7. Egelman DM, Montague PR. Calcium dynamics in the extracellular space of mammalian neural tissue. *Biophysical journal* **76**, 1856-1867 (1999).
8. Cerina M, *et al.* Thalamic Kv 7 channels: pharmacological properties and activity control during noxious signal processing. *British journal of pharmacology* **172**, 3126-3140 (2015).
9. Deleuze C, *et al.* T-type calcium channels consolidate tonic action potential output of thalamic neurons to neocortex. *The Journal of neuroscience : the official journal of the Society for Neuroscience* **32**, 12228-12236 (2012).

10. Ehling P, *et al.* Ca²⁺-dependent large conductance K⁺ currents in thalamocortical relay neurons of different rat strains. *Pflügers Archiv : European journal of physiology* **465**, 469-480 (2013).
11. Huh Y, Cho J. Discrete pattern of burst stimulation in the ventrobasal thalamus for anti-nociception. *PloS one* **8**, e67655 (2013).
12. Huh Y, Cho J. Urethane anesthesia depresses activities of thalamocortical neurons and alters its response to nociception in terms of dual firing modes. *Frontiers in behavioral neuroscience* **7**, 141 (2013).
13. Schoenmakers TJ, Visser GJ, Flik G, Theuvenet AP. CHELATOR: an improved method for computing metal ion concentrations in physiological solutions. *Biotechniques* **12**, 870-874, 876-879 (1992).
14. Helmchen F, Borst JG, Sakmann B. Calcium dynamics associated with a single action potential in a CNS presynaptic terminal. *Biophysical journal* **72**, 1458-1471 (1997).
15. Berridge MJ, Lipp P, Bootman MD. The versatility and universality of calcium signalling. *Nature reviews Molecular cell biology* **1**, 11-21 (2000).
16. Billig GM, Pal B, Fidzinski P, Jentsch TJ. Ca²⁺-activated Cl⁻ currents are dispensable for olfaction. *Nature neuroscience* **14**, 763-769 (2011).
17. Huntsman MM, Porcello DM, Homanics GE, DeLorey TM, Huguenard JR. Reciprocal inhibitory connections and network synchrony in the mammalian thalamus. *Science* **283**, 541-543 (1999).
18. Pifferi S, Dibattista M, Menini A. TMEM16B induces chloride currents activated by calcium in mammalian cells. *Pflügers Archiv : European journal of physiology* **458**, 1023-1038 (2009).
19. Ferrera L, *et al.* Regulation of TMEM16A chloride channel properties by alternative splicing. *The Journal of biological chemistry* **284**, 33360-33368 (2009).

20. Vocke K, *et al.* Calmodulin-dependent activation and inactivation of anoctamin calcium-gated chloride channels. *The Journal of general physiology* **142**, 381-404 (2013).
21. Berkefeld H, Fakler B, Schulte U. Ca²⁺-activated K⁺ channels: from protein complexes to function. *Physiol Rev* **90**, 1437-1459 (2010).

Responses to Reviewer3

Reviewers' comments:

Reviewer #3 (Remarks to the Author):

The manuscript by Ha et al describes some generally well conducted experiments that conclude that the modulation of action potential firing frequency is due to the activation of the calcium activated chloride channel ANO2. This finding is of significant interest as cation channels (e.g. calcium-activated potassium and HCN channels) typically modulate firing frequency, and this is the first demonstration that a chloride channels plays such a role in thalamic neurons. The manuscript is well written and it usually easy to follow the logic of each experiment. In general the data appear to be of good quality and the conclusions sound.

I have some, mostly minor, comments and suggestions.

A: We appreciate the reviewer for the kind and insightful comments on our manuscript and constructive suggestions to strengthen the manuscript. We tried our best to answer all inquiries and concerns raised by the reviewer and edited the figures and manuscript to address these suggestions. The edited text in the revised manuscript is outlined in blue for clarity. Our specific responses to each point raised by the reviewers are also outlined below in blue for clarity.

1. Details should be added to indicate the extent of the region infected with AAV. This will be important for interpretation of the behavioral experiments, as spread of the virus could weaken the conclusion that the behavioral differences resulted from knockdown of ANO2 in thalamic neurons. The inset in Figure 8a showing mCherry staining is not described or discussed in the results text. The percentage of thalamic neurons expressing mCherry should be given with an indication of the numbers/percentage of ANO2-expressing cells infected by the virus.

A. We thank the reviewer for the critical comment to allow us to improve our data. In response to the reviewer, we added supplementary Fig. S7 (also shown below), displaying individual examples for the extent of the AAV-infected region and injected site for each animal measured by the post-hoc staining and modified the text to mention the portion of the AAV-infected area (mCherry-expressing area) to total VB region. We confirmed that the viral infection area was covered by 70.2 ± 3.9 % of VB region, including both the VPL and VPM. We excluded mice whose AAV-infected VB region is below 50%. We added this result to the revised manuscript (Page 15 line 18).

2. It would be instructive to determine the behavioral effect of ANO2 knockdown on an acute pain stimulus (e.g. hot plate or tail flick) that has a short duration where repetitive neuronal firing may be less evident.

A. We respectfully agree with the reviewer on the point that it would be instructive to compare the effect of thalamic ANO2 in the acute type of pain with visceral pain which is categorized to persistent pain. Therefore, in response to the reviewer, we conducted a hot plate test, one of the acute pain tests, in AAV-Scr and AAV-shANO2 mice. In the results, there was no difference in the withdrawal latency of paws from hot plate (shown below, added to Supplementary Fig. S8). We added this point to the main text that the firing frequency adaption by ANO2 channels would function in the persistent type of pain, but be less evident in the acute type of pain. (Page 16 line 6) We thank the reviewer for the constructive suggestion to let us strengthen our manuscript.

3. Standard controls should be used for the immunohistochemistry - staining should be shown for no ANO2 primary antibody (in e.g. Figure 6c). How specific is the antibody staining for neurons? Are there data with this antibody from ANO2-null mice?

A. In response to the reviewer, we added the immunohistochemistry result by using no primary antibody as standard control in the revised supplementary Fig.S5.

In addition, we checked the antibody specificity through western blot by using ANO2-overexpressing and normal NIH/3T3 cells (main Fig.6). Also, the immunohistochemistry data on figure 6d-e indicated that AAV-shANO2-infected TC neurons have lower intensity of ANO2 than AAV-Scr-infected TC neurons, confirming the specificity of ANO2 antibody. This antibody was provided from the research group which made the knockout mice and tested this antibody on ANO2 knockout mice¹.

4. Some of the IHC staining is difficult to see (e.g. Best1). Can the images be improved?

A. In response to the reviewer's suggestion, we replaced the BEST1 IHC staining images with new improved images. These new images were added to the revised manuscript (revised figure 5a).

5. Was the effect of removing calcium from the external solution on spike frequency reversible (page 5/6, Fig 1a) or is this an irreversible effect?

A. Yes, it was fully reversible (data not shown).

6. As niflumic acid can act on some cation channels the conclusion that the NFA inhibited current is an anion current (page 7) is premature. This is only confirmed when the reversal potentials are measured (page 8)

A. We agree with the reviewer on the reviewer's comment that NFA is not selective blocker. To respond to the reviewer's comment, we revised the text in that the NFA-sensitive anion current was replaced with NFA-sensitive current. We also mentioned that this current was identified as an anion current because of the reversal potential. The rephrased sentence was added to the revised manuscript (Page 7 line 20, 22 and Page 8 line 6).

7. The statement (page 17) that ANO2 is localized to the soma of thalamic neurons, unlike SK channels, needs clarification. Does this mean that ANO2 is not expressed on dendrites? ANO2 has been reported to be targeted to dendrites in Purkinje neurons (Zhang et al 2015 PLoS One 10: e0142160).

A. We acknowledged the reviewer's comment on the location of ANO2 in TC neurons. Unlike our report prior to revision, at least, our immunohistochemical data showed the expression of ANO2 strongly localized to the soma of TC neurons. Fig. 6d shows that strong ANO2 signals are localized to the soma of TC neurons and a little bit in the very proximal part of dendrites, but not strong dendritic staining. However, we cannot exclude the possibility that we could not see the strong ANO2 signals in dendritic shape in these images due to the spreading morphology of dendrites of TC neurons compared to Purkinje neurons. We fully agree that it is difficult to conclude that ANO2 is not expressed on dendrites with regard to our data. Therefore, we toned down the statement to "appears to be" in the revised text as the reviewer concerned. (Page 18 line 9)

8. The hyperpolarizing protocol used for Figure 7k should be described in the methods (hyperpolarized to what membrane potential for what time period). At present there is a comment in

the figure legend about hyperpolarizing to $\sim -80\text{mV}$ but the results section mentions 'the hyperpolarizing protocol).

A. We apologize to the reviewer for insufficient information of methods. To respond to the reviewer's comment, we added a brief description on the hyperpolarizing protocol used in the revised figure legend (Page 43 line 8).

9. What is the free calcium concentration in the internal solution with 5Ca-EGTA-NMDG?

A. After calculation using Ca^{2+} -EGTA calculator (<http://maxchelator.stanford.edu/CaEGTA-TS.htm>, pH = 7.3, 25 C), we obtained the free Ca^{2+} concentration in the internal solution with Ca-EGTA-NMDG which was about $30.6\ \mu\text{M}$.

10. State the supplier for the Mini Analysis software.

A. We purchased the Mini Analysis software from Synaptosoft in USA. We included the supplier of the Mini Analysis software on page 22 line 20.

11. A GFAP antibody was used but details are not given in the methods section.

A. We apologize to the reviewer for lack of details and providing inaccurate information. To identify astrocytes, in fact, we used GFAP-GFP mice for immunohistochemistry data confirming mBEST1 expression (revised figure 5a). All the labels as GFAP were actually GFP signals. We corrected this confusion in the revised manuscript and the information about GFAP-GFP mice is added to the methods section (Page 21 line 17).

12. Define mANO1 and mANO2 as mouse ANO1 and ANO2

A. In the revised manuscript, we have included a description of mANO1 and mANO2 on page 11 line 18.

13. Define APA when first introduced (page 8)

A. In response to the reviewer, we decided not to use APA for the abbreviation of apamin to avoid confusion. We replaced APA to apamin in the overall text.

14. Page 18. "...actively emits Cl^- ." Extrudes or effluxes would be a more appropriate word than 'emit'

A. In response to the reviewer, we changed the word to "effluxes" on page 18 line 21.

References

1. Billig GM, Pal B, Fidzinski P, Jentsch TJ. Ca²⁺-activated Cl⁻ currents are dispensable for olfaction. *Nature neuroscience* **14**, 763-769 (2011).

Reviewers' comments:

Reviewer #1 (Remarks to the Author):

In the revised ms, the authors have addressed all my concerns, including the addition of novel data obtained by means of gramicidin patch-clamp which I suggested as an option to further strengthen their study.

I have only one comment:

Unfortunately, I completely missed the fact that the experiments have been carried out at room temperature (see Referee 2 comments), probably because the recording temperature was not mentioned at all in the original ms. Now, the authors state that this information has been provided on p. 22, line 13. However, I cannot find this statement.

I would strongly suggest that the authors would repeat some of their key experiments at a physiologically relevant temperature in vitro, around 33-34 C. The phenomena described in the present work involve a number of temperature-dependent cellular /molecular mechanisms. Moreover, a solution of the present kind, buffered with 26 HCO₃/5% CO₂, has a significantly lower pH at room temperature than at a more physiological one, which may have an effect in itself. I am confident that the authors are able to identify a suitable set of experiments to look at the possible effects of recording temperature.

Minor:

- page 21, line 18: Subheading with "acutely dissociated brain slices": please delete the words "acutely dissociated".

Reviewer #2 (Remarks to the Author):

The manuscript was adequately revised. I have no more concerns.

Reviewer #3 (Remarks to the Author):

The authors have responded positively and modified the manuscript in response to the reviewers comments.

The questions and points that I initially raised have been addressed to my satisfaction. In my view the manuscript has been significantly strengthened by the additional experiments and revisions made by the authors.

Reponses to Reviews

Reviewers' comments:

Reviewer #1 (Remarks to the Author):

In the revised ms, the authors have addressed all my concerns, including the addition of novel data obtained by means of gramicidin patch-clamp which I suggested as an option to further strengthen their study .

A: We thank the reviewer for positive comments on our manuscript and for constructive suggestions to strengthen the manuscript. The edited text in the revised manuscript is outlined in blue for clarity. We tried our best to answer all inquiries and concerns that were raised by the reviewer and added new data to address the suggestions. Our specific responses to each point that was raised by the reviewers are also outlined below in blue for clarity.

Responses to reviewer #1's comments

I have only one comment:

Unfortunately, I completely missed the fact that the experiments have been carried out at room temperature (see Referee 2 comments), probably because the recording temperature was not mentioned at all in the original ms. Now, the authors state that this information has been provided on p. 22, line 13. However, I cannot find this statement.”

A: We apologize for our mistake of missing the temperature information. The recording temperature is now added in p21, line 21 in the revised version.

I would strongly suggest that the authors would repeat some of their key experiments at a physiologically relevant temperature *in vitro*, around 33-34 C. The phenomena described in the present work involve a number of temperature-dependent cellular /molecular mechanisms.”

A: We understand the reviewer's concern and performed an additional experiment to confirm that Ca²⁺ dependence of spike-frequency adaptation also occurs at 32°C and added to Supplementary Fig. S1c,d in the revised manuscript. We also mentioned this in the main text (Page5 line16). As shown below, spike-frequency adaptation was decreased with replacement of Ca²⁺-free extracellular buffer at 32°C which is almost identical to the pattern recorded at 25°C. We wish that these experiments would answer the concerns raised by the reviewer.

It is classically known that the conductance of ion channels are relatively temperature insensitive, with Q_{10} (defined as 10-degree temperature coefficient) of $\sim 1.2-1.5$, from the Hodgkin and Huxley's experiments (Hodgkin et al., 1952; Frankenhaeuser and Moore 1963; Beam and Donaldson 1983; Schwarz 1986; Milburn et al. 1995)^{1, 2, 3, 4, 5}. What this means is that the difference between conductance of a voltage gated sodium channel (for example) at 35°C and 25°C is only 1.2~1.5 fold. This fact is well described in page 50-52 of the book "Ion Channels of Excitable Membranes" by Dr. Bertil Hille which is the most-respected textbook in the subject of ion channel. In fact, in his original work in 1937⁶, Hodgkin proved that action potential is NOT mediated by temperature dependent enzymes by demonstrating the spread of action potential beyond a region of nerve blocked locally by cold. From this classical experiment, Hodgkin demonstrated that action potentials propagate electrically. Based on these classical studies, many electrophysiologists record the firing pattern of neurons and channel conductance at room temperature and can still publish without much of criticism. On the other hand, if one needs to record synaptic activities or study synaptic plasticity, then one must record synaptic activities at 32-34 °C. This is because synaptic transmission and plasticity requires numerous enzymatic reactions including protein synthesis. Even from our own group, there are numerous publications containing the channel recordings at room temperature (*J Neurosci.* 2008 Dec 3;28(49):13331-40; *J Neurosci.* 2011 Jan 26;31(4):1213-8; *PNAS* 2013 Dec 10;110(50):20266-71; *J Neurosci.* 2009 Oct 14;29(41):13063-73; *Science.* 2010 Nov 5;330(6005):790-6; *Cell.* 2012 Sep 28;151(1):25-40; *Nat Commun.* 2014 Feb 5;5:3227)^{7, 8, 9, 10, 11, 12, 13}.

However, to answer the concern raised by the reviewer, we performed the experiments to confirm this idea. We appreciate the reviewer for raising a critical question and wish that these experiments would answer the concerns raised by the reviewer.

Moreover, a solution of the present kind, buffered with 26 HCO₃/5% CO₂, has a significantly lower pH at room temperature than at a more physiological one, which may have an effect in itself. I am confident that the authors are able to identify a suitable set of experiments to look at the possible effects of recording temperature."

A: In response to reviewer's comment, we immediately measured the pH of our aCSF (saturated with at 95% O₂/5% CO₂) at 25°C and 32°C. We found that the pH of our aCSF was 7.30-7.45 at both 25°C

and 32°C. We did not find a significantly lower pH at room temperature. In addition, ANO2 is reported to be pH insensitive. Moreover, the additional experiments to Ca²⁺ dependence of frequency adaptation at 32°C confirmed that firing pattern of TC neuron was not temperature-dependent.

“- page 21, line 18: Subheading with “acutely dissociated brain slices”: please delete the words “acutely dissociated”.”

A:In response to the reviewer's comment, we will delete the words “acutely dissociated.”

References

1. Hodgkin AL, Huxley AF. The components of membrane conductance in the giant axon of Loligo. *The Journal of physiology* **116**, 473-496 (1952).
2. Frankenhaeuser B, Moore LE. The Effect of Temperature on the Sodium and Potassium Permeability Changes in Myelinated Nerve Fibres of *Xenopus Laevis*. *The Journal of physiology* **169**, 431-437 (1963).
3. Beam KG, Donaldson PL. A quantitative study of potassium channel kinetics in rat skeletal muscle from 1 to 37 degrees C. *The Journal of general physiology* **81**, 485-512 (1983).
4. Schwarz JR. The effect of temperature on Na currents in rat myelinated nerve fibres. *Pflugers Archiv : European journal of physiology* **406**, 397-404 (1986).
5. Milburn T, Saint DA, Chung SH. The temperature dependence of conductance of the sodium channel: implications for mechanisms of ion permeation. *Receptors Channels* **3**, 201-211 (1995).
6. Hodgkin AL. Evidence for electrical transmission in nerve: Part II. *The Journal of physiology* **90**, 211-232 (1937).
7. Cheong E, Lee S, Choi BJ, Sun M, Lee CJ, Shin HS. Tuning thalamic firing modes via simultaneous modulation of T- and L-type Ca²⁺ channels controls pain sensory gating in the thalamus. *The Journal of neuroscience : the official journal of the Society for Neuroscience* **28**, 13331-13340 (2008).
8. Cheong E, *et al.* Deletion of phospholipase C beta4 in thalamocortical relay nucleus leads to absence seizures. *Proceedings of the National Academy of Sciences of the United States of America* **106**, 21912-21917 (2009).
9. Cheong E, Kim C, Choi BJ, Sun M, Shin HS. Thalamic ryanodine receptors are involved in controlling the tonic firing of thalamocortical neurons and inflammatory pain signal processing. *The Journal of neuroscience : the official journal of the Society for Neuroscience* **31**, 1213-1218 (2011).

10. Lee J, *et al.* Sleep spindles are generated in the absence of T-type calcium channel-mediated low-threshold burst firing of thalamocortical neurons. *Proceedings of the National Academy of Sciences of the United States of America* **110**, 20266-20271 (2013).
11. Lee S, *et al.* Channel-mediated tonic GABA release from glia. *Science* **330**, 790-796 (2010).
12. Woo DH, *et al.* TREK-1 and Best1 channels mediate fast and slow glutamate release in astrocytes upon GPCR activation. *Cell* **151**, 25-40 (2012).
13. Hwang EM, *et al.* A disulphide-linked heterodimer of TWIK-1 and TREK-1 mediates passive conductance in astrocytes. *Nature communications* **5**, 3227 (2014).

REVIEWERS' COMMENTS:

Reviewer #1 (Remarks to the Author):

The authors have addressed my final comments in a satisfactory manner. I have no further criticisms, just some very minor points which can be corrected at a later stage of processing the paper (see below).

I use the opportunity here to state that my concerns on temperature effects were not related to channel function as such (cf. rebuttal and references to the Hodgkin & Huxley work and Bertil Hille's textbook), but rather to the underlying Ca^{2+} -dependent transduction mechanisms which in turn depend on intracellular calcium regulation.

Also, the pH of a CO_2/HCO_3 buffer IS temperature sensitive (because of the increasing solubility of CO_2 with lower temperature), but the range of values measured by the authors (7.30-7.45) falls well within the variability caused by the difference between 25 and 32 C. Anyway, all the above matters have now been dealt with.

Minor points:

- P. 21 "The most of electrophysiological recordings from the brain slices were performed at 25C except the..."; please change to "*Most of* the electrophysiological recordings... except *for* the..."
- p. 22 "aerated with 95% O_2 /5% CO_2 " should be "equilibrated with..."
- p. 25 "The most of data" should be "Most of the data"

Reponses to Reviews

Reviewers' comments:

Reviewer #1 (Remarks to the Author):

The authors have addressed my final comments in a satisfactory manner. I have no further criticisms, just some very minor points which can be corrected at a later stage of processing the paper (see below).

A: We thank the reviewer for positive comments on our manuscript. We deeply appreciate the reviewer for all the constructive suggestions to strengthen the manuscript.

I use the opportunity here to state that my concerns on temperature effects were not related to channel function as such (cf. rebuttal and references to the Hodgkin & Huxley work and Bertil Hille's textbook), but rather to the underlying Ca^{2+} -dependent transduction mechanisms which in turn depend on intracellular calcium regulation.

Also, the pH of a CO_2/HCO_3 buffer IS temperature sensitive (because of the increasing solubility of CO_2 with lower temperature), but the range of values measured by the authors (7.30-7.45) falls well within the variability caused by the difference between 25 and 32 C. Anyway, all the above matters have now been dealt with.

A: We thank the reviewer for critical questions and tried our best to answer all inquiries and concerns that were raised by the reviewer. Our specific responses to each point that was raised by the reviewers are also outlined below in blue for clarity and incorporated to the revised manuscript..

Minor points:

- P. 21 "The most of electrophysiological recordings from the brain slices were performed at 25C except the..."; please change to "*Most of* the electrophysiological recordings... except *for* the..."

A: In response to the reviewer's comment, we revised the text to "Most of the electro...".

- p. 22 "aerated with 95% O_2 /5% CO_2 " should be "equilibrated with..."

A: In response to the reviewer's comment, we revised the text to "equilibrated with ...".

- p. 25 "The most of data" should be "Most of the data"

A: In response to the reviewer's comment, we revised the text to "Most of the data...".